# Structural basis of αE-catenin–F-actin catch bond behavior

Xiao-Ping Xu[1†], Sabine Pokutta[2†], Megan Torres[2], Mark F Swift[1], Dorit Hanein[1,3*], Niels Volkmann[1,3*], William I Weis[2*]

[1]Scintillon Institute, San Diego, United States; [2]Departments of Structural Biology and Molecular & Cellular Physiology, Stanford University School of Medicine, Stanford, United States; [3]Department of Structural Biology and Chemistry, Pasteur Institute, Paris, France

**Abstract** Cell-cell and cell-matrix junctions transmit mechanical forces during tissue morphogenesis and homeostasis. α-Catenin links cell-cell adhesion complexes to the actin cytoskeleton, and mechanical load strengthens its binding to F-actin in a direction-sensitive manner. Specifically, optical trap experiments revealed that force promotes a transition between weak and strong actin-bound states. Here, we describe the cryo-electron microscopy structure of the F-actin-bound αE-catenin actin-binding domain, which in solution forms a five-helix bundle. In the actin-bound structure, the first helix of the bundle dissociates and the remaining four helices and connecting loops rearrange to form the interface with actin. Deletion of the first helix produces strong actin binding in the absence of force, suggesting that the actin-bound structure corresponds to the strong state. Our analysis explains how mechanical force applied to αE-catenin or its homolog vinculin favors the strongly bound state, and the dependence of catch bond strength on the direction of applied force.

**\*For correspondence:**
dorit.hanein@pasteur.fr (DH);
niels.volkmann@pasteur.fr (NV);
weis@stanford.edu (WIW)

[†]These authors contributed equally to this work

**Competing interests:** The authors declare that no competing interests exist.

## Introduction

The development and maintenance of multicellular organisms depends upon specific adhesion between cells. Morphogenetic movements of sheets of cells are driven by changes in the cytoskeleton of individual cells that are linked to adjacent cells by adhesion molecules. Tissue integrity depends upon response of these adhesive structures to external mechanical perturbation (*Guillot and Lecuit, 2013*; *Ladoux and Mège, 2017*). Cell-cell junctions, including the adherens junctions (AJ), transmit mechanical forces between cells. In AJs, the extracellular domains of cadherins mediate homophilic cell-cell contact, and their cytoplasmic domains are linked to the actin cytoskeleton by β-catenin and α-catenin (*Meng and Takeichi, 2009*; *Shapiro and Weis, 2009*). Specifically, β-catenin binds to the cytoplasmic tails of cadherins and to α-catenin; α-catenin binds to β-catenin and to filamentous (F-)actin (*Figure 1A*). This architecture enables forces generated by actomyosin constriction to be transmitted to neighboring cells during morphogenesis, and conversely allows the actin cytoskeleton to respond to external loads. Similarly, in cell-extracellular matrix adhesions, the extracellular domains of integrins bind to components of the extracellular matrix. The cytoplasmic protein talin binds to integrins and to vinculin, a homolog of α-catenin.

α-Catenin appears to be the major sensor of mechanical force in the AJ. Tension on cadherins depends upon actomyosin activity, β-catenin and α-catenin (*Borghi et al., 2012*). As AJs develop, tension placed on α-catenin promotes conformational changes that enable it to bind to its paralog vinculin (*Barrick et al., 2018*; *Kim et al., 2016*; *le Duc et al., 2010*; *Li et al., 2015*; *Maki et al., 2016*; *Maki et al., 2018*; *Terekhova et al., 2019*; *Yonemura et al., 2010*), whose actin-binding activity further strengthens the cytoskeletal linkage (*Thomas et al., 2013*). In the mature AJ, actin bundles of mixed polarity run parallel to the junction (*Hirokawa et al., 1983*) and lie in close

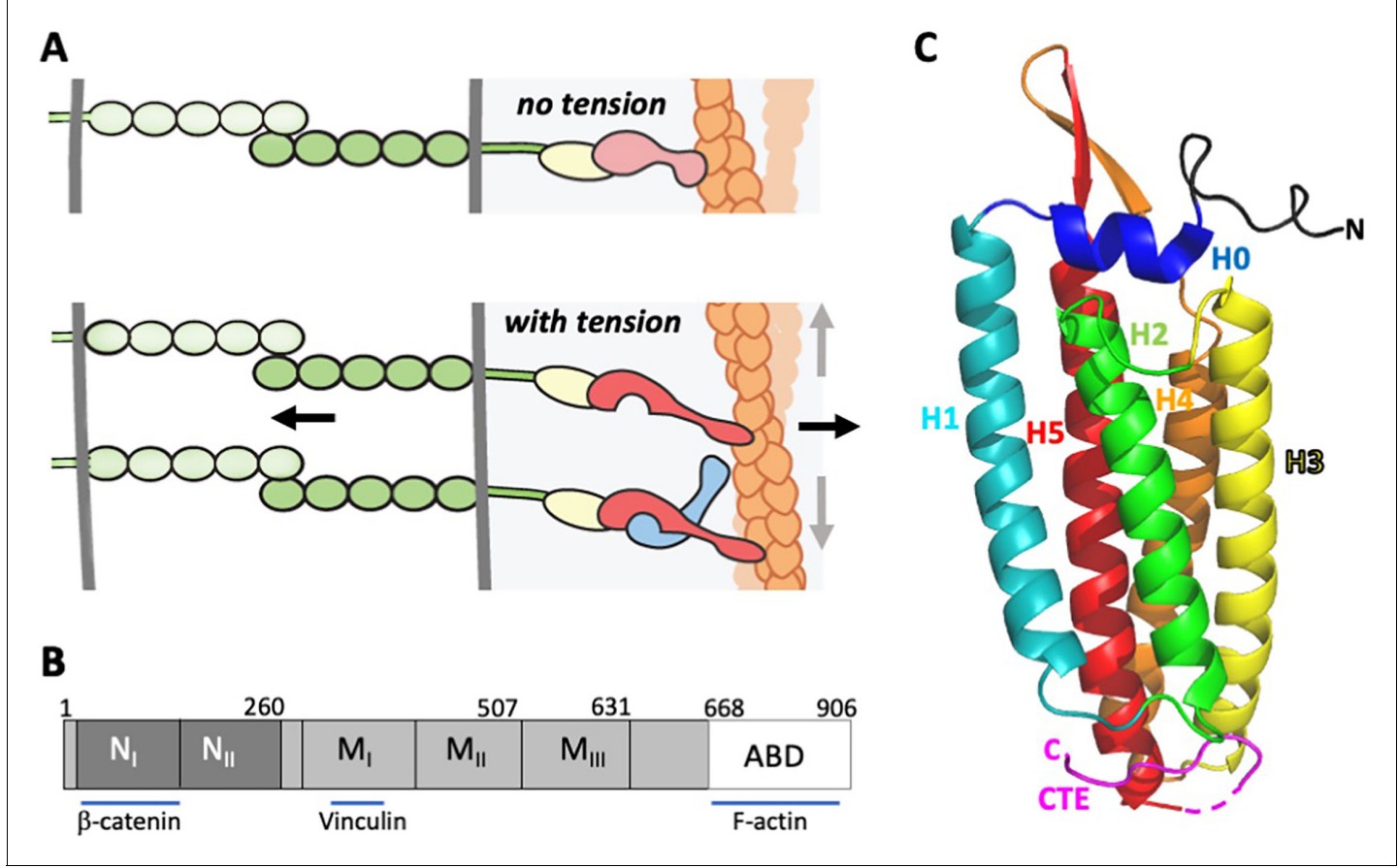

**Figure 1.** α-Catenin in adherens junctions. (**A**) Schematic of AJ and the role of αE-catenin in the connection to the actin cytoskeleton. The extracellular region of cadherins (green) bind to one another between cells, and their cytoplasmic domains bind to β-catenin (yellow). β-Catenin binds to α-catenin (pink/red), which binds to F-actin (orange) weakly in the absence of force (top panel). Tension (indicated by arrows) favors the strong actin-binding state of α-catenin, and also produces conformational changes in α-catenin that lead to recruitment of vinculin (light blue). While the net direction of the force is likely perpendicular to the junction (black arrows), there will be local force components along the mixed-polarity filaments toward their pointed (-) ends through actomyosin contractility (grey arrows). (**B**) Primary structure of αE-catenin; binding sites for β-catenin, vinculin and F-actin are indicated. (**C**) Crystal structure of αE-catenin ABD (*Ishiyama et al., 2018*) (PDB 6dv1); the five helices H1-H5, the N-terminal capping helix H0, and the C-terminal extension (CTE) are labeled.

apposition to the membrane (*Buckley et al., 2014*). Single molecule force measurements of the cadherin–catenin complex binding to actin that employed an optical trap setup revealed that the complex displays catch bond behavior, wherein the interaction with actin is strengthened under mechanical load (*Buckley et al., 2014*). This property was observed subsequently in α-catenin and full-length vinculin themselves, as well as the actin-binding domain (ABD) of vinculin, indicating that the homologous ABDs of these proteins confer catch bond behavior (*Abore et al., 2020*; *Huang et al., 2017*). In both proteins, the catch bond is asymmetric: force directed toward the (-) end of the actin filament results in a longer lived bond than when force is directed toward the (+) end (*Abore et al., 2020*; *Huang et al., 2017*).

α-Catenin has three major domains: an N-terminal β-catenin-binding domain, a middle (M) domain, followed by a flexible linker to the C-terminal ABD (*Figure 1B*). The three-dimensional structures of the ABD from αE(epithelial)- and αN(neuronal)-catenins have been determined (*Ishiyama et al., 2018*; *Ishiyama et al., 2013*), and the ABD has also been visualized in the crystal structure of a nearly full-length αE-catenin (*Rangarajan and Izard, 2013*). These structures reveal that the ABD comprises a bundle of five helices, preceded by a short N-terminal helix (designated H0) that sits on top of the bundle (*Figure 1C*). Helices 2, 3, 4 and 5 (H2-5) form an antiparallel four-helix bundle in which hydrophobic residues from each helix contribute to a hydrophobic core. Helix 1 interacts with the side of the four-helix H2-H5 bundle. A long C-terminal extension (CTE), residues

844–906, follows H5. In different structures, the CTE adopts different conformations and is partly disordered. The vinculin ABD likewise adopts a similar five helix bundle architecture, albeit with a shorter H1 and no H0 (*Bakolitsa et al., 2004*; *Bakolitsa et al., 1999*; *Borgon et al., 2004*).

In the optical trap data, the distribution of bound lifetimes of the cadherin/catenin complex or vinculin at any particular force follows a bi-exponential distribution, indicating that there are two distinct actin-bound states, weak and strong (*Buckley et al., 2014*; *Huang et al., 2017*). The population of longer lifetimes increases with force, and modeling of these data indicated that the catch bond behavior arises because force enhances interconversion of the weakly- to the strongly bound state (*Buckley et al., 2014*). These observations explain why binding of the cadherin–catenin complex to actin in solution, that is, under no external load, is weak; force shifts the equilibrium between the weakly and strongly bound states and thereby produces tighter binding (*Buckley et al., 2014*; *Yamada et al., 2005*).

While the catch bonding of αE-catenin and vinculin to actin is established, to date there has been no molecular explanation of how force changes the structure of their ABDs to promote strong binding. Recent work in solution demonstrated removal of H0 from the ABD of αE-catenin enhances its affinity for F-actin, suggesting that force-dependent removal of this structural element is important for catch bonding (*Ishiyama et al., 2018*). However, vinculin lacks H0 yet also displays catch bond behavior (*Bakolitsa et al., 2004*; *Bakolitsa et al., 1999*; *Borgon et al., 2004*; *Huang et al., 2017*). Here, we present the structure of the αE-catenin ABD lacking H0 bound to F-actin obtained by cryo-electron microscopy (cryo-EM). The structures of the free (*Ishiyama et al., 2018*; *Ishiyama et al., 2013*; *Rangarajan and Izard, 2013*) and the actin-bound forms of the complete αE-catenin ABD, as well as the structures of the vinculin ABD in the absence (*Bakolitsa et al., 2004*; *Bakolitsa et al., 1999*; *Borgon et al., 2004*; *Mei et al., 2020*) or presence (*Mei et al., 2020*) of actin provide an explanation for the weak to strong actin-binding transition, and biochemical and mutational data support this model.

## Results

### Structure of αE-catenin ABD bound to F-actin

To understand the αE-catenin-actin filament interaction in molecular detail, we obtained a three-dimensional cryo-EM reconstruction of ADP-actin filaments bound to a truncated αE-catenin ABD (residues 671–906) at 3.6 Å resolution (*Figure 2*, *Figure 2—figure supplement 1*, *Table 1*), which we reported at lower resolution previously (*Hansen et al., 2013*). This construct deletes the first half of, and thereby destabilizes, the short H0, and binds 4.5x more strongly than the complete ABD (residues 666–906) (*Table 2*, *Figure 2—figure supplement 2*; *Ishiyama et al., 2018*). Consistent with the previously described cooperative binding by this construct observed in TIRF and cryo-EM (*Hansen et al., 2013*), we observed either bare actin filaments or stretches of filaments continuously bound by αE-ABD added at 10 μM (*Figure 2—figure supplement 1A*). At the same concentration, we were unable to observe binding of the complete ABD to actin filaments in the electron microscope (*Figure 2—figure supplement 1B*). Given the $K_D$ values of these two constructs (*Table 2*), small changes in concentration likely have a large effect on actin decoration. Indeed, a comparable structure using an ABD construct spanning residues 664–906 was produced using 20 μM ABD (*Mei et al., 2020*).

The cryo-EM reconstruction allowed us to build an atomic model of the complex (*Figure 2*), using the structures of bare ADP actin filaments (*Chou and Pollard, 2019*) and the αE-ABD crystal structure (*Ishiyama et al., 2018*) as starting points. The EM map was poor in the αE-catenin CTE, and we were able to correct a sequence registration error in our first model based on the coordinates of the 3.2 Å resolution structure reported by *Mei et al., 2020* provided by Dr. G Alushin. Only the first three and last residues of actin could not be placed with confidence. For the αE-catenin ABD, there was no detectable density for residues 671–698, or from 872 to 906. In addition, six residues in the loop connecting H4 and H5 could not be modeled. Local resolution analysis shows that the most well-defined region is within the actin filament core and gradually falls off toward larger radii (*Figure 2—figure supplement 1C*). Importantly, the interface of the ABD and F-actin is well defined in the cryo-EM map, as is the conformation of the helical bundle.

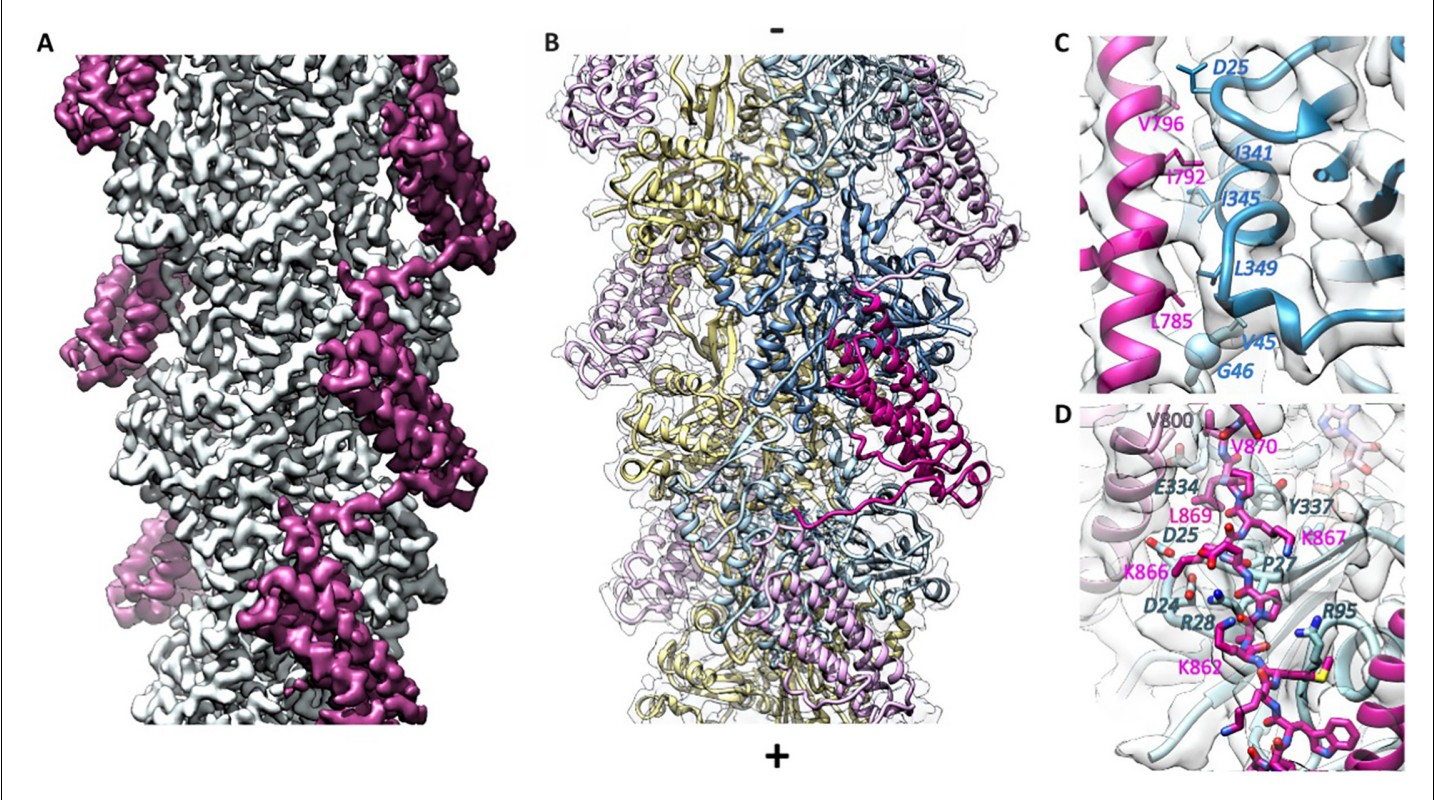

**Figure 2.** Cryo-EM analysis of the αE-catenin ABD–F-actin complex. (A) Cryo-EM map of the actin-ABD structure. The segmented ABDs are shown in magenta. The (-) end of the filament is shown at the top, and the (+) end at the bottom. (B) Molecular model of a section of an actin filament bound to αE-catenin ABDs, same orientation as (A) and with transparent density map overlaid. Actin protomers are colored according to their long-pitch helix in blue and yellow. The bound ABDs are shown in magenta and pink. (C, D) Closeups of model and cryo-EM map showing residues on H4 (panel C) and the CTE (panel D) that have been studied by site-directed mutagenesis. The ABD is shown in red, and two monomers of actin in different shades of blue. In (D), a neighboring copy of the ABD along the filament is shown in pink. Actin residue labels are *italicized*.

The online version of this article includes the following figure supplement(s) for figure 2:

**Figure supplement 1.** Cryo EM analysis.

**Figure supplement 2.** Representative gels and binding curves for actin co-sedimentation assays with αE-catenin N-terminal deletion constructs.

**Figure supplement 3.** Close-up of αE-catenin–F-actin interactions.

**Figure supplement 4.** Representative gels and binding curves for actin co-sedimentation assays with αE-catenin C-terminal deletion constructs.

**Figure supplement 5.** Alignment of α-catenin and vinculin ABD sequences.

The helical rise (27.4 Å) and twist (−166.9°) of F-actin in the reconstruction are practically identical to those of bare F-actin (*Chou and Pollard, 2019*; *Merino et al., 2018*), indicating that the ABD does not induce any major changes into the filament. Consequently, the root-mean square deviation between bare ADP actin filaments and actin with the ABD bound is low (0.76 Å, Cα deviations). The most notable difference is in the conformation of the subdomain two loop (D-loop), a region implicated in changes associated with the ATP hydrolysis cycle (*Chou and Pollard, 2019*; *Merino et al., 2018*), stiffness and stability (*Kang et al., 2012*; *Pospich et al., 2017*), and in filament disassembly (*Grintsevich et al., 2017*). In the present structure, this region is in a 'closed' conformation similar to that observed in the bare ADP-bound F-actin structures (*Chou and Pollard, 2019*; *Merino et al., 2018*). Modeling suggests that the alternative 'open' D-loop conformation that occurs in equilibrium with the 'closed' conformation in other nucleotide states (*Merino et al., 2020*) may clash with the bound αE-catenin. Relative to the bare ADP-actin structure, however, D-loop residues 45–50 move significantly, with M47 showing the largest displacement of about 5 Å. This region contacts the αE-catenin ABD. It has been noted that tensile forces imposed on actin by the thin ice needed for cryo-EM imaging may affect the D-loop (*Galkin et al., 2012*), and we have previously observed similar differences at low resolution upon αE-catenin binding (*Hansen et al., 2013*), but whether tension has

**Table 1.** Cryo-EM data collection and model statistics.

**Data collection**

| | |
|---|---|
| **Microscope** | **Titan Krios** |
| Voltage [kV] | 300 |
| Detector | Falcon II |
| Magnification | 75,000 |
| *Exposure parameters* | |
| Total dose [e⁻/Å²] ($e^-/\text{Å}^2$) | 60 |
| Exposure time [s] | 1.0 |
| Pixel size [Å] | 1.035 |
| Defocus range [μm] | −0.8 to −2.8 |
| **Data processing** | |
| Images used | 4769 |
| Initial segments | 728,331 |
| Final segments | 422,822 |
| *Helical symmetry* | |
| Rise [Å] | 27.4 |
| Twist [°] | −166.9 |
| Resolution [Å] | 3.6 |
| FSC threshold | 0.143 |
| Sharpening B-factor [Å²] | −96 |
| **Refinement** | |
| Initial models [PDB IDs] | 6djo, 6dv1 |
| Non-hydrogen atoms | 25,248 |
| Model resolution [Å] | 3.5 |
| FSC threshold | 0.5 |
| *RMS deviations* | |
| Bond lengths [Å] | 0.007 |
| Bond angles [°] | 0.92 |
| Rotamer outliers [%] | 4.4 |
| Mean B-factor [Å²] | 70.4 |
| **Validation** | |
| Molprobity score | 2.08 |
| Clash score | 6.28 |
| *Ramachandran plot* | |
| Favored [%] | 96.3 |
| Allowed [%] | 3.7 |
| Disallowed [%] | 0.0 |
| CC (mask) | 0.86 |
| CC (volume) | 0.82 |
| EMringer score | 2.29 |

any role in the conformation observed here is unclear given the direct contacts with the ABD. Moreover, the refinement procedure used to generate high-resolution structures from cryo-EM images selects and enforces a single conformation (see Methods), so it is likely that in order to achieve the highest resolution reconstruction possible, other information content including alternative conformations, was lost.

**Table 2.** Affinities of αE-catenin ABD constructs for actin, determined by co-sedimentation.
$K_D$ values and standard deviations for αE-catenin 666–906, 671–906, and 692–906 are the average of three replicate measurements. $K_D$ values for the other constructs are the average of two measurements. For αE-catenin 671–906 W859A binding did not reach saturation and therefore only a lower limit for the $K_D$ is given. N.D., no detectable binding. Representative binding curves and corresponding gels are shown in **Figure 2—figure supplements 2** and **4**.

| αE-catenin ABD variant | $K_D$ (µM) (SD) |
| --- | --- |
| 666–906 (full length) | 8.5 (0.7) |
| 671–906 | 2.0 (0.3) |
| 692–906 | 0.7 (0.3) |
| 696–906 | 0.4 (0.05) |
| 699–906 | 0.5 (0.1) |
| 671–872 | 5.0 (0.1) |
| 671–868 | N.D. |
| 671–864 | N.D. |
| 699–868 | 2.8 (0.1) |
| 671–906 W859A | >35 |

In contrast to the local changes in F-actin, the ABD undergoes large structural rearrangements upon complex formation. Compared to the unbound ABD crystal structure, the N-terminus through the last turn of H1 is disordered (**Figure 3A**). We note that an F-actin-bound αE-catenin ABD structure has been reported recently for the complete ABD (664-906) (**Mei et al., 2020**), and the same residues are disordered, demonstrating that the truncation of H0 in our construct has no influence on the actin-bound structure. The remaining four helices rearrange to bind to the filament. A key part of this change involves the long loop that connects H4 and H5, where the first strand of a β-hairpin becomes a two-turn extension of H4 (residues 795–801) that forms contacts with actin (**Figures 2C** and **3A,B**).

In addition to the changes in the helical bundle, the remaining turn of H1 moves up from the bottom of the rest of the helical bundle (**Figure 3** and **Figure 3—figure supplement 1**). This moves W705 (H1) away from an aromatic cluster formed with Y837 (H5), and W859 (CTE), and shifts the CTE upward such that M861 of the CTE now packs with Y837 and W859 (**Figure 3—figure supplement 1**). The CTE past 862, which is disordered in the isolated structure, forms an extended peptide that interacts with actin (**Figure 2D**). The backbone carbonyl oxygens of αE-catenin K862 and P864 form hydrogen bonds with actin residues R28 and R95 (**Figure 2—figure supplement 3A**). αE-catenin K866 forms salt bridges with actin D24 and D25, K867 forms a hydrogen bond with the backbone carbonyl of R28 and contacts V30, and L869 packs against actin residues A26, P27, E334 and Y337 (**Figure 2—figure supplement 3A**). The importance of the change in the CTE and formation of contacts with F-actin is highlighted by the effect of mutating W859 to alanine, which lowers the affinity for actin approximately 10-fold (**Table 2**, **Figure 2—figure supplement 4**). Finally, we note that the CTE sits between actin and the next ABD along the long pitch of the filament, and V870 packs against V800 at the top of H4 of the neighboring ABD. This interaction may contribute to the cooperative binding of the ABD to actin (**Hansen et al., 2013**).

The actin-bound αE-catenin ABD complex structure is consistent with previously reported mutations that weaken its binding to F-actin. **Ishiyama et al., 2018** mutated H4 residues L785, I792 and V796 to alanine. L785 had the most severe effect, reducing the $K_D$ ~15 x; it interacts at the interface of two longitudinally adjacent actin monomers, including L349 of one monomer and V45 and G46 in the D-loop of the other (**Figure 2C**). I792A and V796A also had significant effects on affinity; I792 forms a packing interaction with I345 of actin and V796 packs against actin residues I341 and D25 (**Figure 2C**). **Chen et al., 2015** found several point mutants that severely weakened binding, including I792A. K842A eliminates a hydrogen bonds with H87 and Y91 of actin (**Figure 2—figure supplement 3B**); and K866A eliminates salt bridges with actin D24 and D25 (**Figure 2—figure supplement 3A**). **Pappas and Rimm, 2006** removed sets of positively charged residues and saw

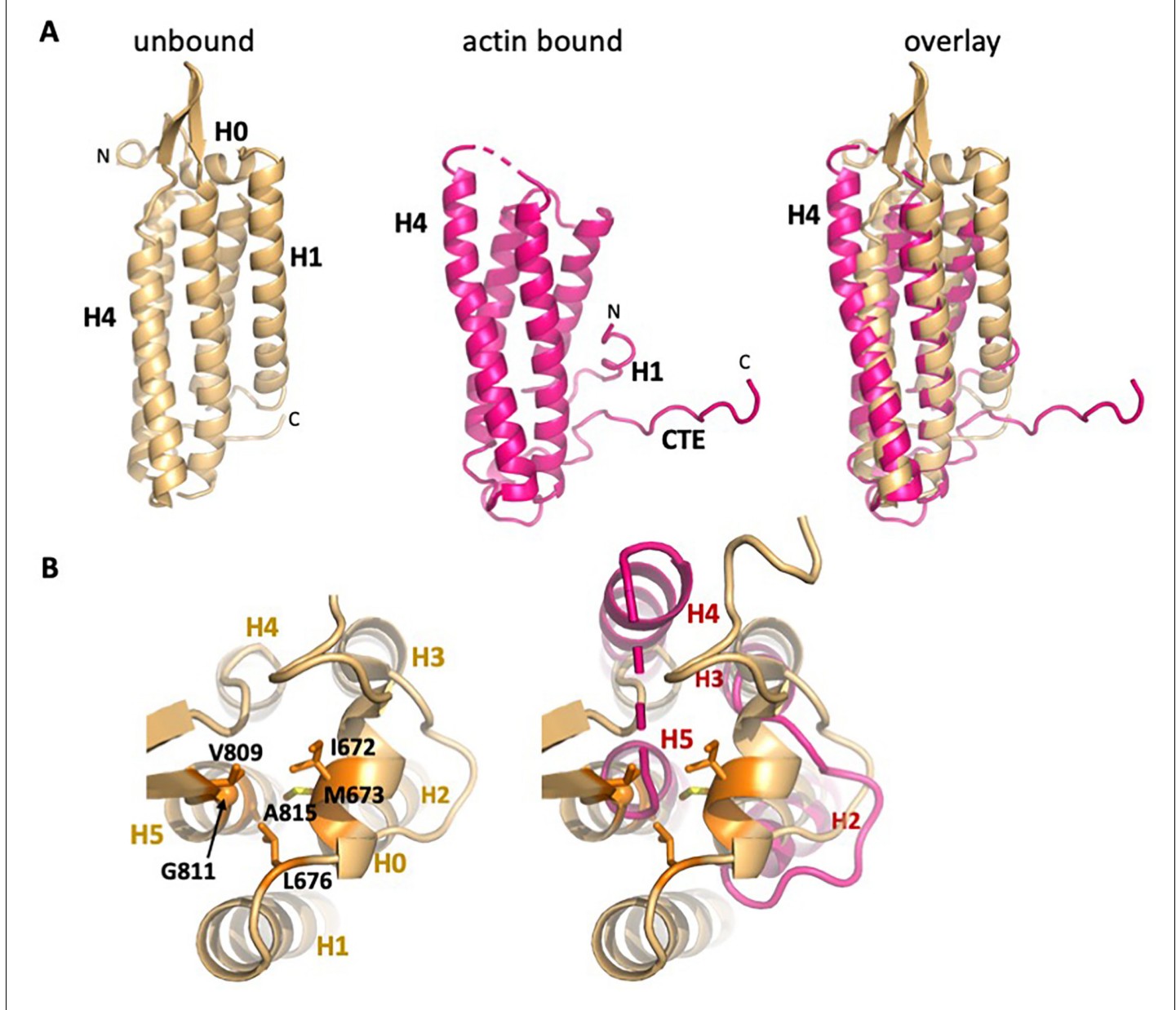

**Figure 3.** Overall changes in ABD structure upon binding to F-actin. Comparison of the unbound ABD crystal structure (PDB 6dv1; light orange) with the ABD in the actin-bound state (magenta). (A) Overall comparison; the orientation is rotated approximately 180° from that shown in *Figure 1c*. (B) Top view of H0 packing interactions lost upon its removal, and rearrangements of helices 2–5. The left panel depicts packing interactions of H0 residues I672, M673 and L676 with H5 residues V809, G811 and A815 (all highlighted in orange), and the right panel overlay shows the resulting changes in H4 and H5.

The online version of this article includes the following figure supplement(s) for figure 3:

**Figure supplement 1.** Differences in the CTE free and bound to F-actin.

**Figure supplement 2.** Comparison of free and actin-bound αE-catenin and vinculin ABDs.

only modest effects on binding; the largest effect was the triple mutant K747A/K748A/K797A; K797 packs into the hydrophobic core of the bundle and forms a salt bridge with actin E334, whereas the other two lysine residues point into solvent on the other side of the domain (*Figure 2—figure supplement 3C*).

Deletion of the C-terminal αE-catenin residues 865–906 compromises actin binding (*Pokutta et al., 2002*), whereas a construct ending at 883 binds (*Chen et al., 2015*; *Pappas and Rimm, 2006*). This observation is consistent with the contacts observed between actin and residues 866–869 (*Figure 2D*, *Figure 2—figure supplement 3A*). To more precisely determine which residues of the αE-catenin CTE observed to contact actin are critical for binding, we prepared a series of C-terminal truncations of the αE-catenin 671–906 construct and compared their affinities (*Table 2*, *Figure 2—figure supplement 4*). We confirmed that removing residues 865–906 produced no detectable binding. Removal of residues 869–906 eliminated detectable F-actin binding from the variant starting at 671, and when these residues are removed from the higher affinity variant lacking H1 (starting at 699), binding is detectable but 5.6x weaker. These findings confirm the contributions of L869 and V870 to binding (*Figure 2—figure supplement 3A*). Surprisingly, there is a slight loss of affinity upon removal of residues 873–906, even though we do not observe these residues in the structure. *Mei et al., 2020* proposed that these C-terminal residues absent in the structure may mediate a small increase in affinity when actin is placed under tension. However, deleting these residues weakens the affinity (albeit slightly) in a solution assay, which indicates that they have a role independent of tension. It is possible that these are highly dynamic interactions that are not sufficiently stable to be visualized in the cryo-EM structure.

The structural changes and interactions with actin observed here appear to be conserved throughout the α-catenin/vinculin family. Specifically, the residues that mediate the interactions with actin are strongly conserved throughout the α-catenin sequences (*Figure 2—figure supplement 5*). Moreover, although relatively few of the actin-contacting residues are conserved in vinculin (notably, those in the C-terminal portion of H4), the vinculin ABD undergoes a similar structural transition upon binding to actin (*Figure 3—figure supplement 2*; *Kim et al., 2016*; *Mei et al., 2020*).

## H0 and H1 regulate actin affinity

Despite the large changes between the free and F-actin-bound ABD structures, we see no evidence for multiple conformations of the ABD when bound to F-actin, suggesting that the four-helix state is the stably bound one. To assess whether the rearranged state is significantly populated in solution, we compared the proteolytic sensitivity of the H0-deleted ABD used in the EM structure (671-906) in the presence and absence of F-actin, using the protease elastase. We found that in isolation the domain was resistant to digestion, whereas binding to F-actin led to the appearance of smaller, protease-resistant fragments (*Figure 4A*). N-terminal sequencing of the SDS-PAGE bands corresponding to these fragments revealed cleavage of H1 at residues A689 and S703 (*Figure 4B*). The cleavage at S703 is consistent with the very weak cryo-EM density observed between residues 699 and 702, which likely indicates that this turn of helix is flexible. Moreover, the cleavage at 689 suggests that H1 becomes unstructured and flexible when dissociated from the H2-5 bundle. The resistance of H1 to protease in the absence of F-actin implies that its association with the H2 and H5 surface is strongly favored in solution. Helix 1 of the vinculin actin-binding domain, which forms a similar four helix bundle when bound to actin (*Figure 3—figure supplement 2*; *Kim et al., 2016*; *Mei et al., 2020*), also becomes proteolytically sensitive upon binding to actin (*Bakolitsa et al., 1999*). The proteolysis data from both αE-catenin and vinculin suggest that the free energy of binding to actin drives the structural rearrangement of the ABD.

If the association of H0 and H1 with the four-helix bundle inhibits the rearrangement of the structure to a stable actin-bound conformation, they should weaken the affinity of the ABD for actin. Therefore, we prepared a series of αE-catenin ABD variants in which H0 and H1 were deleted or truncated. As more N-terminal sequence was deleted, the affinity became stronger, such that deleting H1 through residue 698 results in approximately 18x stronger binding to actin filaments relative to the complete ABD (*Table 2*, *Figure 2—figure supplement 2*). Notably, Ishiyama et al. deleted H0 only and saw about a ~ 3 x increase in affinity (*Ishiyama et al., 2018*), similar to change we observed when H0 is disrupted by deleting residues 666–670 (*Table 2*, *Figure 2—figure supplement 2*). This observation is also consistent with the observation that the crystal structure of the αN-catenin ABD lacking H0 still forms the 5-helix assembly observed in the complete ABD (*Ishiyama et al., 2018*). Overall, the deletion data indicate that removal of H0 and most of H1 produce a strong actin-binding species. Notably, the enhanced binding conferred by deleting H1 can compensate for the loss of residues 869–906 (*Table 2*; compare 671–868 and 699–868).

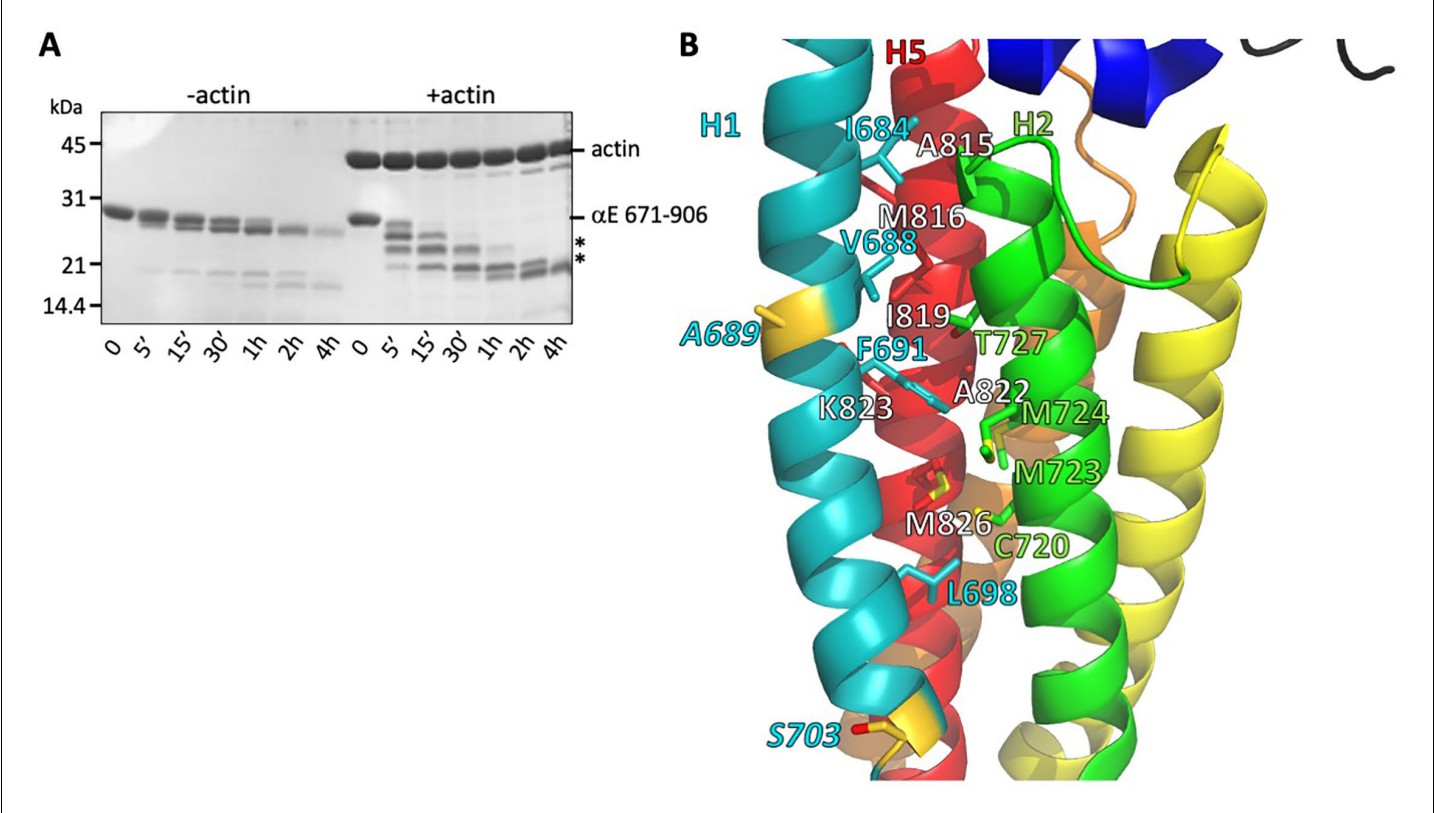

**Figure 4.** Stability of the H1- H2/H5 interface. (A) Time course of elastase digestion of αE-catenin 671–906. The two smaller fragments analyzed by N-terminal sequencing are indicated with asterisks. (B) Rainbow diagram of the unbound αE-catenin ABD (PDB 6dv1), colored as in *Figure 1c*. Residue labels for the H5 (red) helix are shown in white for clarity. The two residues at the elastase cut sites are indicated in gold. Side chains in the H1- H2/H5 interface are shown in stick representation.

Sequence features of the ABD support the idea that the free energy of binding to actin drives dissociation of H0 and H1 and rearrangement of the remaining helices. H0 has three conserved hydrophobic residues (I672, M673 and L676) that pack against three residues at the N-terminal region of H5 (*Figure 3B*), two of which (V809 and G811) are poorly ordered in the actin-bound structure. H1 binds to the outer face of the H2–H5 four helix bundle, interacting with a surface formed by H2 and H5. Several hydrophobic H1 residues (I684, V688, F691 and L698) are buried in this interface (*Figure 4B*), which would disfavor dissociation of H1. However, the H1 interaction surface formed by H2 and H5 is not strongly hydrophobic, comprising four methionine residues (M723, M724, M816 and M826), C720, T727, I819, A822 and the aliphatic portion of K823 (*Figure 4B*), which suggests that there would not be a large destabilization of the four-helix bundle upon removing H1 from this surface. Indeed, the construct starting at 699, which deletes all of the H1 sequence missing in the EM structure, is well behaved in solution (it is monomeric as assayed by size exclusion chromatography-coupled multi-angle light scattering; data not shown), consistent with the idea that exposure of this surface is not energetically disfavored. Notably, the mildly hydrophobic character of this H2/H5 surface is strongly conserved throughout the α-catenin family (*Figure 2—figure supplement 5*).

### Insights into catch bond mechanism

Our structural and biochemical data indicate that removal of H0 and H1 enable the structural transition of the C-terminal half of H4 and movement of the CTE (*Figure 3*), which result in additional contacts with F-actin and stable binding (*Figures 2* and *5*). Given that the vinculin ABD lacks H0 but its structure bound to actin shows the same four-helix, rearranged bundle relative to vinculin in solution (*Mei et al., 2020*), and that the crystal structure of the αN-catenin ABD lacking H0 retains the five-helix bundle architecture of the full ABD (PDB 6duw, 6duy) (*Ishiyama et al., 2018*), it is clear that

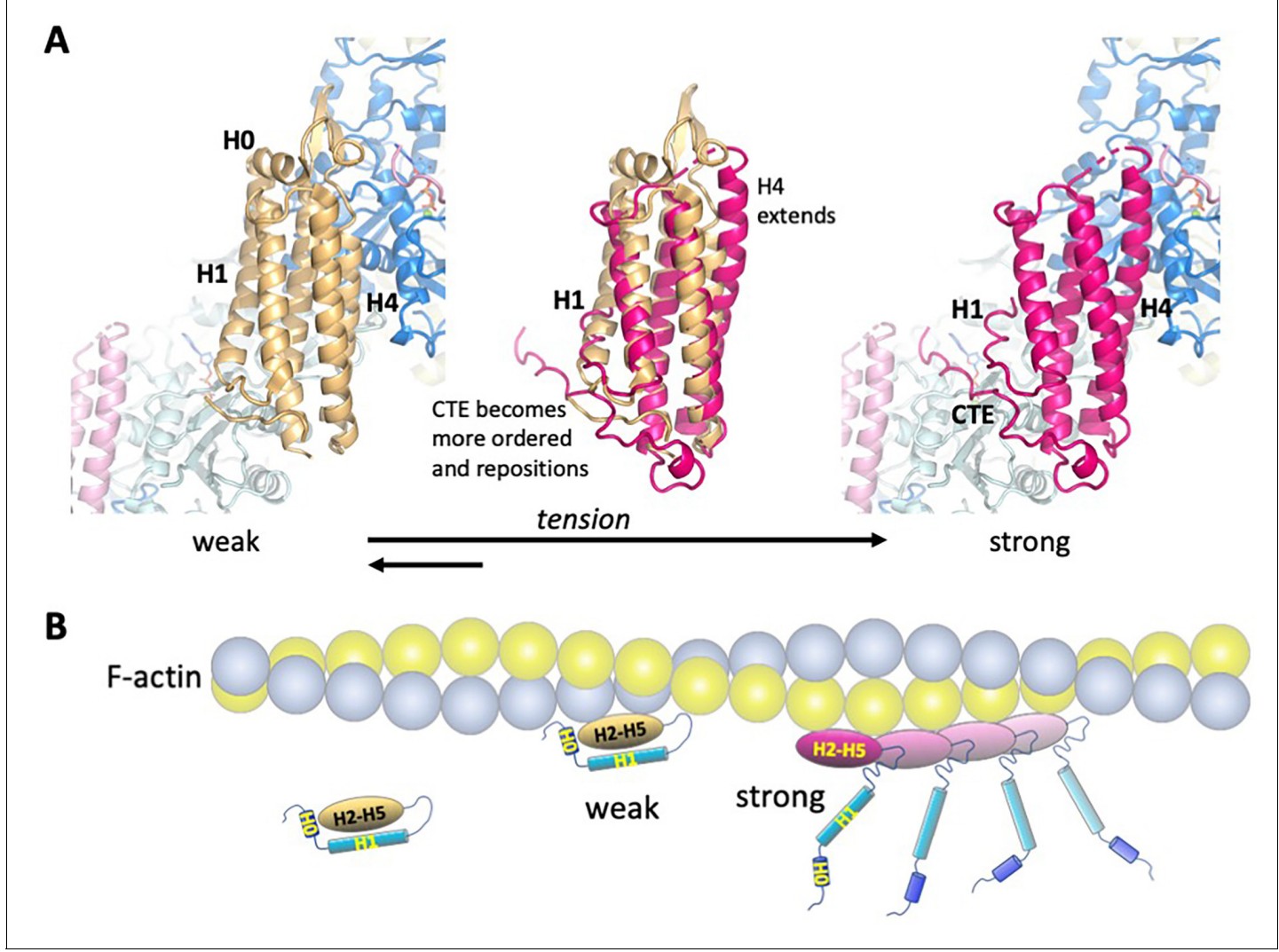

**Figure 5.** Model of the weak and strong actin-binding states of αE-catenin. (**A**) Superposition of the isolated αE-catenin ABD on the actin-bound structure reveals no major clashes with F-actin (left panel). When the ABD is bound to F-actin, H0 and H1 dissociate from the H2-5 bundle, which results in the extension and shift of the C-terminal part of H4 as well as ordering and repositioning of the CTE to bind to actin. (**B**) Schematic diagram of αE-catenin ABD conformational states when unbound, weakly bound, and strongly bound to actin. Cooperative binding of the ABD, as observed in the cryo-EM structure, is illustrated for the strong state. Note that in the strong state, the H0 and H1 regions are drawn as helices when dissociated from the H2-5 bundle, but it is likely that they are unstructured in this case (see text for details).

The online version of this article includes the following figure supplement(s) for figure 5:

**Figure supplement 1.** Clash of isolated ABD structure with actin.

removal of H1 is the major determinant in achieving the high-affinity actin-binding state. The binding data in solution, that is, in the absence of applied force, demonstrate that the free energy of binding to actin drives this transition. Optical trap data from both αE-catenin and vinculin indicate that force enhances the stability (specifically, the bound lifetime) of the ABD-actin interaction (*Abore et al., 2020*; *Buckley et al., 2014*; *Huang et al., 2017*). Modeling of the optical trap data indicated that the major effect of force is to prevent the transition from a strongly bound to a weakly bound state (*Buckley et al., 2014*; *Huang et al., 2017*). Force also promotes the transition to a strongly bound state, although this is a smaller effect (*Buckley et al., 2014*; *Huang et al., 2017*) that is consistent with the binding energy to actin driving the rearrangement to the strong state. Thus, while force is not needed to stably bind actin, it enhances the strength of the ABD-actin interaction by shifting the equilibrium between weakly and strongly bound states toward the strong state.

To gain structural insight into how force affects the weak and strong actin-binding states of the ABD, we superimposed the isolated αE-catenin ABD structure (PDB 6dv1) on the actin-bound version (*Figure 5A*, *Figure 5—figure supplement 1*). This revealed clashes of actin residues K328, I330 and P333 (*Figure 5—figure supplement 1*) with ABD residues D813 and M816 in the first turn of H5, as well as E799 in the H4-H5 connection and K683 of H1. Modeling indicates that a few other minor clashes can be alleviated by changes of side chain rotamers. Given the proteolytic digestion data suggesting that H1 dynamically associates with the H2-H5 region (*Figure 4*) and that crystal structures of the αE- and αN-catenin ABDs in isolation show variability in the H4-H5 connection (*Ishiyama et al., 2018*), it seems likely that small changes (on the order of 1 Å) in this region could accommodate the actin surface without the wholesale conformational changes that produce the four-helix conformation. Moreover, as the N-terminal half of H4 and almost all of H5 of the isolated and actin-bound ABD structures superimpose closely (*Figure 5A*, middle), it is likely that in the five-helix conformation these regions could form interactions similar to those visualized in the EM structure. However, key interactions made by the C-terminal part of H4, including those of I792 and V796 (*Figure 2C*), would not form. These observations suggest that with small changes of H1 and the first turn of H5, the five-helix conformation could bind F-actin weakly, and we propose that this conformation represents the weak binding state observed in the optical trap (*Figure 5B*).

Assuming that the four-helix conformation observed in the complex with F-actin represents the strong state, it is likely that force on the ABD prevents H1 and H0 from associating with the rest of the ABD (*Figure 6A*, *Video 1*). Tension on the N-terminus of the ABD that is stably bound to F-actin will prevent re-association of H1 and H0, thereby favoring the strongly bound state and enhancing its lifetime. Conversely, if the five-helix conformation binds weakly, force would provide additional energy to drive the dissociation of H0 and H1 from the bundle (*Figure 6B*, *Video 2*), thereby facilitating the transition to the strongly bound state. The dissociation of H1 also enables the ordering of the CTE, which contributes to high-affinity binding (*Table 2*) through its direct interactions with actin and possibly by contributing to cooperative binding through its interaction with the longitudinal ABD neighbor on actin (*Figure 2*, *Figure 2—figure supplement 3A*, *Figure 5B*).

## Discussion

Catch bond behavior has been observed in a number of proteins subject to mechanical force in a variety of biological contexts. Examples include the extracellular portions of cell adhesion molecules such as bacterial FimH that attaches to the urinary tract epithelium, selectins that mediate rolling of leukocytes on endothelia, integrins that mediate cell-extracellular matrix adhesion, and classical cadherins (*Pruitt et al., 2014*). For FimH, selectins and integrins, hinges between domains change position under mechanical load, and these changes can be transmitted to binding domains in a variety of ways to promote a strong ligand-binding state (*Le Trong et al., 2010*; *Pruitt et al., 2014*; *Rakshit et al., 2012*; *Springer, 2009*; *Springer et al., 2008*; *Xiao et al., 2004*). More recent work has revealed catch bonding to F-actin by the intracellular adhesion proteins αE-catenin and vinculin. These proteins also have multiple domains that are likely to change their relative positions upon application of force, as has been demonstrated for αE-catenin (*Barrick et al., 2018*; *Choi et al., 2012*; *Kim et al., 2016*; *Li et al., 2015*; *Terekhova et al., 2019*). Attachment of the α-catenin N-terminal domain to β-catenin and the C-terminal ABD to actin (*Figure 1B*) implies that tension is transmitted through the entire protein.

The structure of the actin-bound αE-catenin ABD presented here provides a molecular-level explanation of the catch bond behavior of αE-catenin as well as the homologous vinculin (*Abore et al., 2020*; *Buckley et al., 2014*; *Huang et al., 2017*). By serving as a bridge between β-catenin and actin, αE-catenin is placed under tension, which likely stretches the linker between the M domain and the ABD and thereby applies tension to the ABD H0 and H1 regions. Likewise, binding of vinculin to talin and actin in focal adhesions will stretch the loop that precedes the ABD. Tension stabilizes the four-helix, strong-binding conformation of these ABDs bound to F-actin by preventing rebinding of H1 (and H0 in the case of αE-catenin) (*Figure 6A*, *Video 1*). The stabilization of the αE-catenin CTE in the four-helix conformation is likely an important component of the strongly bound state: it not only forms interactions with actin but also contacts a neighboring ABD on the filament, which may underlie its cooperative binding to F-actin (*Buckley et al., 2014*; *Hansen et al., 2013*). Moreover, modeling of the isolated αE-catenin ABD structure on actin shows few clashes and

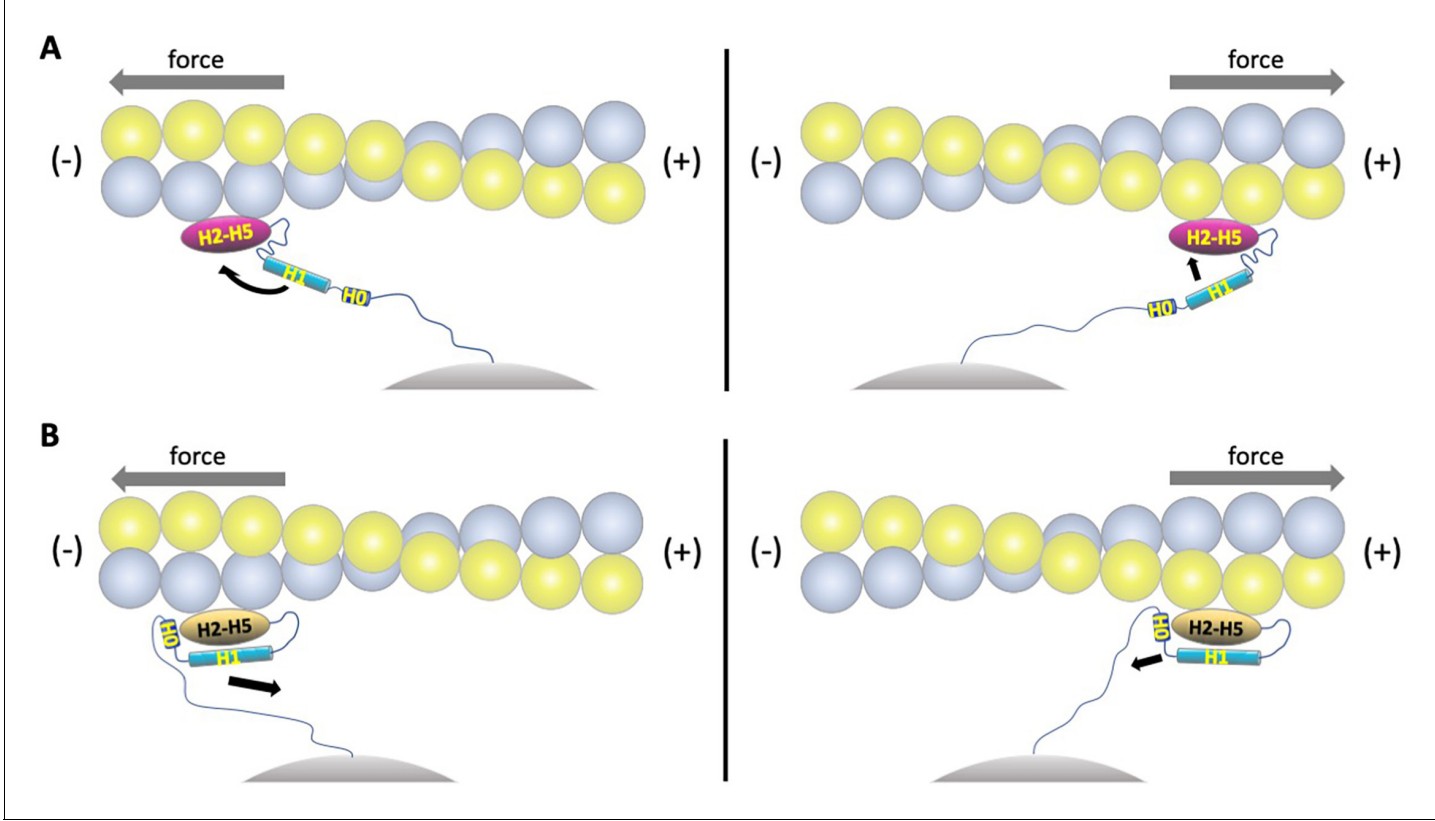

**Figure 6.** Model of directional catch bonding. Actin and the αE-catenin ABD in the strong and weak states are illustrated as in *Figure 5B*. The N-terminus of the ABD is shown tethered to a stationary point, that is, as part of the cadherin/β-catenin/α-catenin complex. The grey arrows indicate the direction of force. (**A**) Tension applied to the bound strong state prevents re-binding of H0/H1 to the H2-H5 bundle. Force applied in the (-) direction will move the H1 sequence away from the H2-H5 bundle and place this region in an unfavorable orientation for rebinding, whereas force directed in the (+) direction will place the H1 sequence closer to and in a more favorable orientation for rebinding. See *Video 1* for an animated version. (**B**) Tension applied to the bound weak state will remove H0/H1 from the H2-H5 bundle. Force applied in the (-) direction will tend to pull H0/H1 away from the H2-H5 bundle, whereas force in the (+) direction is predicted to have a smaller effect on H0/H1 dissociation. See *Video 2* for an animated version.

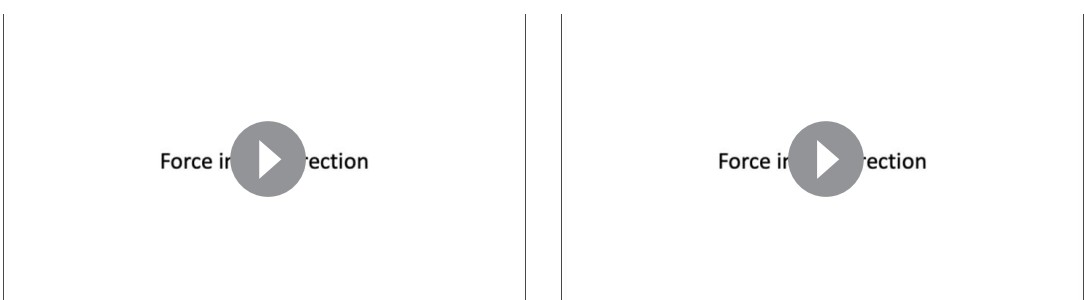

**Video 1.** The strong to weak transition. Animation illustrating changes in the αE-catenin ABD going from the strong to weak actin-bound state. The first half of the movie shows force is applied in the (-) direction, and the second half shows force in the (+) direction, highlighting the difference in distance for reassociation of H0 and H1 in the two directions.
https://elifesciences.org/articles/60878#video1

**Video 2.** The weak to strong transition. Animation illustrating changes in the αE-catenin ABD in going from the weak to strong actin-bound state. The first half of the movie shows force is applied in the (-) direction, and the second half shows force in the (+) direction, highlighting the difference in distance for reassociation of H0 and H1 in the two directions.
https://elifesciences.org/articles/60878#video2

suggests that a five-helix conformation close to that of the isolated structure forms a subset of the interactions observed in the cryo-EM structure, and this likely corresponds to the weakly bound state. If so, then tension applied to the weakly bound state could facilitate dissociation of H0 and H1 and therefore transition to the strongly bound conformation (*Figure 6B*, *Video 2*).

It has been noted that tension applied to an extended peptide can remove this element from another part of the same protein or a partner, and that the mechanical properties of the now flexible, disordered polypeptide contribute to catch bond behavior (*Guo et al., 2019*; *Guo et al., 2018*; *Yuan et al., 2017*). While beyond the scope of this work, it is interesting to consider that the CTE has such a role in the catch bonding of the ABD. In this case, its contribution is difficult to clearly define since its ordering in the presence of actin appears to be intimately coupled to the change in H1. Moreover, the proteolysis data (*Figure 4A*) suggest that H1 becomes disordered in the transition from the strongly to weakly bound state. If H1 (and possibly H0) unfold when detached from the rest of the ABD, it would provide a flexible linker that extends in the direction of force. Future work using molecular dynamics simulations and optical trap studies should allow us to investigate this aspect of the ABD.

Optical trap and biolayer interferometry measurements indicate that in the absence of force, 80–90% of the actin-bound αE-catenin molecules are in the weak state (*Buckley et al., 2014*; *Ishiyama et al., 2018*). The presence of a small fraction in the strong state implies that the free energy of binding to actin promotes dissociation of H1 even in the absence of force (*Figure 6A*). Our deletion data indicate that removal of H1 does not destabilize the rest of the ABD, but the burial of several hydrophobic H1 residues in the H2/H5 interface (*Figure 4B*) suggests that H1 dissociation would be disfavored. The energy input by mechanical force acting over a certain distance helps to overcome this barrier.

A recent study proposed that removing H0 enables the strongly bound, force-enhanced state of αE-catenin (*Ishiyama et al., 2018*), although the structure of the actin-bound ABD showing the dissociated H1 and rearranged H2-H5 region was not available to these authors. Their proposal was based in part on the observation that in order to fit biolayer interferometry F-actin binding data for the complete ABD, a model invoking two species with different affinities was needed (*Ishiyama et al., 2018*). The increased affinity without H0 (*Ishiyama et al., 2018*; *Table 2*) shows that the stabilization energy provided by the interaction of H0 with the rest of the molecule contributes to the barrier to the transition to the stably bound conformation in αE-catenin, as H0 likely interferes with the rearrangements of H2–H5 (*Figure 3B*). However, the fact that the crystal structure of a mutant αN-catenin ABD lacking H0 is not significantly different from the crystal structure of the complete ABD (*Ishiyama et al., 2018*) indicates that removing only H0 does not produce the rearranged, strong-binding four-helix bundle conformation in solution. Moreover, the vinculin ABD lacks H0, yet also forms catch bonds with actin and forms a four-helix bundle similar to that of αE-catenin when bound to F-actin (*Huang et al., 2017*; *Kim et al., 2016*; *Mei et al., 2020*). Also, both the vinculin ABD and αE-catenin ABD lacking H0 are resistant to protease digestion when not bound to actin (*Figure 4*; *Bakolitsa et al., 1999*). Finally, removal of H0 and H1 produces a significantly higher affinity for F-actin than removing only H0 (*Table 2*). These observations imply that dissociation of H1 is the major determinant of stable binding to actin. The free energy of binding to F-actin drives H1 dissociation, and mechanical force further enhances the dissociated state. Although it is not apparently integral to the catch bond behavior, we speculate that the conserved H0 of α-catenin serves to tune the stability and force response of the ABD; in the disordered state bound to actin, it will provide additional flexible elements that may affect the detailed catch bond behavior.

The catch bonds formed by αE-catenin, vinculin and their ABDs with actin show a strong asymmetry (*Abore et al., 2020*; *Huang et al., 2017*) (N. Bax, D. Huang, A. Wang, A. Dunn, and W.I.W., manuscript in preparation). Specifically, force directed toward the (-) end of the filament greatly enhances the strongly bound, long-lived state, whereas force toward the (+) end has a more modest effect on bound lifetimes. (We originally reported [*Buckley et al., 2014*] that the catch bonding of the cadherin/β-catenin/αE-catenin complex was not asymmetric, but this proved to be due to limitations in the sensitivity of the instrument used in that study; N. Bax, D. Huang, A. Wang, A. Dunn, and W.I.W., manuscript in preparation). The lifetimes of the weak and strong actin-bound states observed in the optical trap were described by a modified Bell model (*Bell, 1978*; *Evans, 2001*) in which the transition state energy depends on the force acting over the distance from the ground to transition state (*Huang et al., 2017*). The dependence of bound lifetime on force was the same for

force directed toward either the (+) or (-) end of the filament, implying that there is a different distance to the transition state in the two directions (*Huang et al., 2017*). Although we do not have a detailed molecular model for the transition between weak and strong states, we note that the force vector experienced by α-catenin or vinculin is not necessarily aligned with the actin filament and will depend on the way the molecule is tethered. Thus, with the rest of α-catenin or vinculin bound to a stationary anchor (i.e. cadherin/β-catenin or integrin/talin), force in the (-) direction of the filament would result in pulling the N-terminus of the ABD in the opposite direction, positioning H1 away from the H2-H5 bundle and disfavoring its rebinding (*Figure 6A*, *Video 1*). Likewise, force applied to the weakly bound five-helix conformation would result in H1 'peeling off' from the bundle (*Figure 6B*, *Video 2*). Force applied in the opposite (+) direction would tend to move the N-terminus of the ABD such that H1 would be more aligned with its orientation found in the five-helix state, giving it a higher probability of rebinding to the H2-H5 bundle (*Figure 6A*, *Video 1*), yielding a smaller effect of force in stabilizing the strong state in this direction. While myosin II contractility also applies force in the (-) direction of actin filaments, the actual direction of the applied force vectors for both αE-catenin and vinculin depends on the detailed geometry of the full adhesive complexes and their organization in the cell. Determining these parameters will be needed to fully understand catch bonding by these proteins.

# Materials and methods

## Key resources table

| Reagent type (species) or resource | Designation | Source or reference | Identifiers | Additional information |
|---|---|---|---|---|
| Recombinant DNA reagent | pGEX-TEV | *Choi et al., 2012* https://doi.org/10.1074/jbc.M511338200 | | Ampicillin resistance; expression in bacterial cultures; pGEX-KG plasmid (ATCC) with a new TEV protease site Contact Weis lab for distribution |
| Recombinant DNA reagent | pGEX-4T-3 | GE Healthcare | 28-9545-52 | Vector for thrombin-cleavable GST fusion protein expression in bacteria |
| Strain, strain background (*Escherichia coli*) | BL21 (DE3) Codon-Plus RIL | Agilent | 230245 | Strain for expressing recombinant proteins |
| Software, algorithm | RELION3 3.0.8 | *He and Scheres, 2017* https://doi.org/10.1016/j.jsb.2017.02.003 *Scheres, 2012* https://doi.org/10.1016/j.jmb.2011.11.010 *Zivanov et al., 2018* https://doi.org/10.7554/eLife.42166 | RRID:SCR_016274 | |
| Software, algorithm | MotionCor2 1.3.0 | *Zheng et al., 2017* https://doi.org/10.1038/nmeth.4193 | RRID:SCR_016499 | |
| Software, algorithm | Gctf 1.06 | *Zhang, 2016* https://doi.org/10.1016/j.jsb.2015.11.003 | RRID:SCR_016500 | |
| Software, algorithm | CTFFIND4 4.1.5 | *Rohou and Grigorieff, 2015* https://doi.org/10.1016/j.jsb.2015.08.008 | RRID:SCR_016732 | |

*Continued on next page*

*Continued*

| Reagent type (species) or resource | Designation | Source or reference | Identifiers | Additional information |
|---|---|---|---|---|
| Software, algorithm | pyCoAn 0.3.0 | *Volkmann and Hanein, 1999* https://doi.org/10.1006/jsbi.1998.4074 | Revision 1419 | |
| Software, algorithm | Phenix 1.17.1 | *Afonine et al., 2018* https://doi.org/10.1107/S2059798318006551 | RRID:SCR_014224 | |
| Software, algorithm | Coot 0.8.9 | *Emsley et al., 2010* https://doi.org/10.1107/S0907444910007493 | RRID:SCR_014222 | |
| Software, algorithm | EMRinger | *Barad et al., 2015* https://doi.org/10.1038/nmeth.3541 | Via Phenix | |
| Software, algorithm | MolProbity | *Chen et al., 2010* https://doi.org/10.1107/S0907444909042073 | RRID:SCR_014226 Via Phenix | |
| Software, algorithm | ResMap 1.14 | *Kucukelbir et al., 2014* https://doi.org/10.1038/nmeth.2727 | | |
| Software, algorithm | SBGrid | *Morin et al., 2013* https://doi.org/10.7554/eLife.01456 | RRID:SCR_003511 | |
| Software, algorithm | UCSF Chimera 1.14 | *Pettersen et al., 2004* https://doi.org/10.1002/jcc.20084 | RRID:SCR_004097 | |
| Software, algorithm | GraphPad Prism 8.0.2 | GraphPad Software, Inc | Version 263 RRID:SCR_002798 | |

## Expression and purification of αE-catenin ABD constructs

αE-catenin ABD constructs were cloned into a pGEX-4T-3 or pGEX-TEV bacterial expression vector; the latter is a modified pGEX-KG vector with a tobacco etch virus (TEV) protease recognition site inserted after the GST-tag and the thrombin cleavage site. There are four additional amino acids are left at the N-terminus after Thrombin or TEV cleavage (GSPN in case of the pGEX-4T3 vector and GGIL in case of the pGEX-TEV vector). N-terminal GST-fusion proteins were expressed in *E. coli* BL21cells. Cells were grown at 37°C to an $OD_{600}$ of 0.8–1.0 and induced overnight at 18°C with 0.5 mM isopropyl-1-thio-β-d-galactopyranoside. Cells were harvested by centrifugation and pellets were resuspended in 20 mM Tris pH 8.0, 200 mM NaCl and 1 mM DTT. Before lysis in an Emusliflex (Avastin), protease inhibitor cocktail (Mixture Set V, Calbiochem) and DNase (Sigma) were added. After centrifugation at 38,500 × *g* for 30 min, the lysate was incubated with glutathione-agarose beads for 1 hr at 4°C. After washing the beads with PBS containing 1 M NaCl and 1 mM DTT, the beads were equilibrated with either 20 mM Tris pH 8.5, 150 mM NaCl, 1 mM DTT for thrombin cleavage or 20 mM Tris pH 8.0, 150 mM NaCl, 1 mM DTT, 1 mM EDTA, 10% glycerol for TEV cleavage. The protein was cleaved for 2 hr at room temp with thrombin or overnight at 4°C with TEV. Cleaved protein was eluted from the beads and further purified on a cation exchange column (Mono S 10/100, GE Healthcare) in MES pH 6.5, 1 mM DTT buffer with a 0–500 mM NaCl gradient, followed by size exclusion chromatography (Superdex S200, GE Healthcare) in 20 mM HEPES pH 8.0, 150 mM NaCl, 1 mM DTT.

## Actin-binding assay

G-actin prepared from rabbit muscle (*Spudich and Watt, 1971*) was stored in 40 µM aliquots at −80°C. Frozen aliquots were thawed on ice and centrifuged for 20 min at 140,717 x *g* in a Beckman TLA 100 rotor. After centrifugation, the concentration was determined by UV absorbance at 290 nm and G-actin was polymerized by addition of 10x F-buffer (100 mM pH 7.5 Tris, 500 mM KCl, 20 mM $MgCl_2$, 10 mM ATP) and incubation for 1 hr at room temperature. Aliquots from the same batch of

actin were used for all polymerization assays, and efficient polymerization was confirmed by pelleting at 20 min at 140,717 x *g* and analysis of the supernatant and pellet by SDS page. F-actin was stored for up to 2 weeks at 4°C. For sedimentation assays, F-actin was diluted to 4 µM with buffer A (20 mM HEPES pH 8.0, 150 mM NaCl, 1 mM DTT, 2 mM $MgCl_2$, 0.5 mM ATP, 1 mM EDTA). A dilution series of purified αE-catenin ABD in 20 mM HEPES pH 8.0, 150 mM NaCl and 1 mM DTT was set up and an equal volume of 4 µM F-actin or buffer A was added. The mixture was incubated for 30 min at room temperature. Samples were centrifuged in a Beckman TLA100 rotor at 140717 x *g* for 20 min at 4°C. The supernatant was carefully removed and the pellet resuspended in reducing Laemmli buffer. Samples were run on SDS PAGE. Coomassie-stained bands were scanned and quantified on a LI-COR Odyssey scanner (LI-COR Biosciences). To extrapolate concentration from band intensity a dilution series of αE-catenin ABD was run in parallel for each assay and stained and destained under the same conditions as the assay itself. To correct for SDS-PAGE loading errors, for each concentration of αE-catenin, its band intensity was normalized by the ratio of the actin band intensity at that point and the average actin band intensity calculated over all concentration points. The data were analyzed in the program GraphPad Prism (GraphPad Software, La Jolla, CA,) and fitted with a 'single binding with Hill coefficient' model, with the exception of the αE-catenin 671–906 W859A mutant. In that case, the curves did not reach saturation, and fitting with a Hill coefficient was not possible, so a 'One site-specific binding' model was used to obtain $K_D$ estimates. In this case the binding is sufficiently weak that we report a lower limit on the $K_D$ rather than a specific value (*Table 2*).

## Electron cryo-microscopy sample preparation

Rabbit skeletal actin was prepared as described (*Kang et al., 2012*; *Spudich and Watt, 1971*) and was used within 1 week of preparation. Fresh complete (residues 666–906) or truncated αE-catenin-ABD (residues 671–906) were used within 1–2 days of preparation. Both filamentous actin and the respective αE-catenin ABD construct were diluted into KMEI buffer (10 mM Imidazole pH 7, 50 mM KCl 2 mM $MgCl_2$, 1 mM EGTA, 0.2 mM ATP, 2 mM DTT) at 0.125 mg/ml actin and 0.25 mg/ml ABD, corresponding to an ABD concentration of 10 µM. After 10 min of incubation, 5 or 4 µl from the final 1:2 (wt/wt) mixture was applied to plasma cleaned C-flat copper grids 2/1 or 2/2 (Protochips Inc), respectively. After 1 min of incubation in a humidified chamber, excess liquid was manually blotted, and the samples were plunge-frozen in liquid nitrogen-cooled liquefied ethane using an in-house designed cryo-plunger.

Screening for the best sample mixture ratios and blotting conditions was performed on a Tecnai Spirit T12 electron microscope (ThermoFisher Scientific) equipped with Eagle CMOS imaging device (ThermoFisher Scientific), operated at a voltage of 120 kV and a defocus between −1.5 and −2.5 µm. Micrographs were visually inspected for quality of filaments, filament density, background, and the presence of bound ABD. The choice of samples for data collection was based on evaluation of these parameters. Data sets were acquired on Titan Krios electron microscope (ThermoFisher Scientific) equipped with an XFEG and operated at a voltage of 300 kV. Although the sample preparation protocol was optimized, we had to screen for usable grids and grid squares manually. Images were recorded on a Falcon II direct detection camera (ThermoFisher Scientific) under minimal dose conditions using the automatic data collection software EPU (ThermoFisher Scientific). Within each selected grid hole, two positions were imaged, each with a total exposure of 1 s. A total of 5573 dose-fractionated image stacks with seven frames each were collected with a 1.035 Å pixel size at defoci ranging from −0.8 µm to −2.8 µm in four separate, independent imaging sessions.

## Cryo-EM image processing

Dose weighting and motion correction were applied using MotionCor2 version 1.4.0 (*Li et al., 2013*) using anisotropic motion correction with 5 × 5 patches. The initial defocus was estimated either using Gctf 1.06 (*Zhang, 2016*) or CTFFIND4 version 4.1.5 (*Rohou and Grigorieff, 2015*), depending on the imaging session. 804 images were discarded during real-time screening at data collection time for excessive drift, strong astigmatism, and low visibility of Thon rings. The remainder was processed with the helical reconstruction routines in RELION3 version 3.0.8 (*He and Scheres, 2017*; *Scheres, 2012*; *Zivanov et al., 2018*). Briefly, the helices were divided into overlapping boxes that were essentially treated as individual, independent particles (with modified Bayesian prior accounting for constraints implied by helicity) to allow sorting of the segments into different conformations

and selecting the most well-defined of the conformations present in the sample, a prerequisite for reaching high resolution. For the truncated αE-catenin ABD (671-906) bound to rabbit actin, a total of 728,331 filament segments were extracted using a box size of 200 × 200 pixels from 63,480 manually picked filaments. Two-dimensional reference-free classification for the data set was carried out in RELION3 to eliminate bad segments and segments that showed no evidence for bound ABD, reducing the number of segments from 728,331 to 422,822. An in-house rabbit skeletal actin filament reconstruction filtered to 40 Å resolution was used for an initial model. After several rounds of 3D classification and refinement followed by manual removal of bad particles and further enrichment of segments showing clear decoration, the estimated resolution of the reconstruction, using the 0.143 FSC cutoff gold-standard procedure implemented in RELION3, reached 3.6 Å after postprocessing. The helical rise of the reconstruction was 27.4 Å with a helical twist of $-166.9°$. The postprocessing included RELION3-based CTF refinement, B-factor sharpening, and application of a soft-edged mask generated in RELION3 corrected for helical edge effects using pyCoAn 0.3.0, an extended python version of CoAn (*Volkmann and Hanein, 1999*). Additional sharpening was applied using pyCoAn. The reconstruction was then symmetrized within pyCoAn using the refined helical parameters. Local resolution estimates were calculated with ResMap 1.1.4 (*Kucukelbir et al., 2014*) and RELION3.

A molecular model was produced starting from a structure of bare ADP actin filaments (PDB code 6djo) and the αE-ABD crystal structure (PDB code 6dv1), then iteratively adjusted manually with Coot 0.8.9 (*Emsley et al., 2010*) and subjected to real-space refinement in Phenix 1.17.1 (*Afonine et al., 2018*). Quality indicators, including MolProbity (*Chen et al., 2010*) and EmRinger (*Barad et al., 2015*) scores, were calculated with Phenix. Some of the processing was done in the SBGrid environment (*Morin et al., 2013*). Figures were generated with UCSF Chimera version 1.14 (*Pettersen et al., 2004*).

The coordinates and cryo-EM map of the αE-catenin–F-actin complex have been deposited in the Protein Data Bank, identifiers 6WVT and EMD-21925, respectively.

## Limited proteolysis

Limited proteolysis of αE-catenin 671–906 was performed with elastase (Worthington Biochemical). αE-catenin ABD (8 µM) was incubated in the presence or absence of 8 µM F-actin for 1 hr at room temperature in 5 mM Tris pH 8.0, 50 mM potassium chloride, 2 mM magnesium chloride and 0.5 mM DTT. After addition of elastase to a final concentration of 0.009 mg/ml, aliquots were removed after 5', 15', 30', 1 hr, 2 hr, 4 hr and the proteolysis reaction was stopped by addition of SDS sample buffer and boiling. Samples were run on SDS–PAGE and gels were stained with Coomassie Blue. For N-terminal sequencing, bands were transferred on a PVDF membrane. Bands of ~24 and ~26 kDa found only in the presence of F-actin were excised and submitted for N-terminal (Edman) protein sequencing.

## Acknowledgements

We are grateful to Dr. Gregory Alushin for providing coordinates of the αE-catenin ABD bound to actin published in the accompanying paper (*Mei et al., 2020*), which allowed us to correct an error in the CTE of our model during the proof stage of this manuscript. We thank Max Pokutta for technical assistance. WIW thanks Alex Dunn for discussions. This work was supported by US National Institutes of Health grant GM118326 (DH, NV and WIW) and GM131747 (WIW). NIH grants S10-OD012372 and S10-OD026926, and PEW innovation funds 864K625 to DH funded the purchase of the Titan Krios electron cryo-microscope (ThermoFisher Scientific), Falcon two direct detector (ThermoFisher Scientific), and upgrades for both the T12 and Titan Krios hardware and software including the Eagle CMOS imaging device (ThermoFisher Scientific) on the Tecnai Spirit T12.

## Additional information

### Funding

| Funder | Grant reference number | Author |
|---|---|---|
| National Institutes of Health | GM118326 | Dorit Hanein<br>Niels Volkmann<br>William I Weis |
| National Institutes of Health | GM131747 | William I Weis |
| National Institutes of Health | S10-OD012372 | Dorit Hanein |
| National Institutes of Health | S10-OD026926 | Dorit Hanein |
| Pew Charitable Trusts | 864K625 | Dorit Hanein |

The funders had no role in study design, data collection and interpretation, or the decision to submit the work for publication.

### Author contributions

Xiao-Ping Xu, Formal analysis, Investigation, Methodology; Sabine Pokutta, Conceptualization, Formal analysis, Investigation, Writing - review and editing; Megan Torres, Formal analysis, Investigation; Mark F Swift, Investigation, Methodology; Dorit Hanein, Funding acquisition, Investigation, Methodology, Project administration, Writing - review and editing; Niels Volkmann, Conceptualization, Formal analysis, Funding acquisition, Investigation, Methodology, Project administration, Writing - review and editing; William I Weis, Conceptualization, Formal analysis, Funding acquisition, Investigation, Writing - original draft, Project administration, Writing - review and editing

### Author ORCIDs

Dorit Hanein ⓘ https://orcid.org/0000-0002-6072-4946
William I Weis ⓘ https://orcid.org/0000-0002-5583-6150

### Decision letter and Author response

Decision letter https://doi.org/10.7554/eLife.60878.sa1
Author response https://doi.org/10.7554/eLife.60878.sa2

## Additional files

### Supplementary files
• Transparent reporting form

### Data availability

The coordinates and cryo-EM map of the αE-catenin-F-actin complex have been deposited in the Protein Data Bank, identifiers 6WVT and EMD-21925, respectively.

The following datasets were generated:

| Author(s) | Year | Dataset title | Dataset URL | Database and Identifier |
|---|---|---|---|---|
| Xu X-P, Pokutta S, Torres M, Swift MF, Hanein D, Volkmann N, Weis WI | 2020 | Coordinates of the αE-catenin-F-actin complex | https://www.rcsb.org/structure/6WVT | RCSB Protein Data Bank, 6WVT |
| Xu X-P, Pokutta S, Torres M, Swift MF, Hanein D, Volkmann N, Weis WI | 2020 | Cryo-EM map of the αE-catenin-F-actin complex | http://www.ebi.ac.uk/pdbe/entry/emdb/EMD-21925 | Electron Microscopy Data Bank, EMD-21925 |

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
