## [Decision Letter]

Thank you for submitting your article "Structural basis of αE-catenin-F-actin catch bond behavior" for consideration by *eLife*. Your article has been reviewed by three peer reviewers, including Christopher Hill as the Reviewing Editor and Reviewer #1, and the evaluation has been overseen by a Senior Editor. The following individuals involved in review of your submission have agreed to reveal their identity: Jie Yan (Reviewer #2).

The reviewers have discussed the reviews with one another and the Reviewing Editor has drafted this decision to help you prepare a revised submission.

Summary:

A structure is reported for a complex between αE-catenin and F-actin. The catenin construct is a truncated form that promote F-actin binding, apparently by destabilizing an alternative weakly-binding conformation. The actin-bound catenin conformation differs from that of a previously report structure of a longer construct, which presumably displays the autoinhibited weak-binding form. The primary mechanistic insight is a plausible mechanism for the catch-bond behavior of the catenin-actin interaction. The structural-mechanistic model is supported by biochemical data and sequence conservation, and represents an important advance in understanding.

Essential revisions:

Abstract

1) In the Abstract, the authors mentioned "Upon binding to actin, the first helix of the bundle dissociates and the remaining four helices and connecting loops rearrange to form the interface with actin." This conclusion is suggested from the higher proteolytic sensitivity of the H0-deleted ABD in the presence of F-actin, instead of a direct result. However, if the binding to F-actin is enough to dissociate H1, why do the authors propose that there are two populations of the bound ABDs, a weaker bound state where the H1 is not dissociated and a stronger bound where the H1 is displaced, which equilibrium is shifted by the force applied to the ABD? To allow the force play a significant role, it seems to me that in the absence of force, the auto-inhibited weaker species should be the prevalent one, so that the force can modulate the binding by releasing this auto-inhibition. Perhaps it is more reasonable to think that the observed higher proteolytic sensitivity of the H0-deleted ABD in the presence of F-actin indicates that the H1 helix becomes more dynamic, instead of being dissociated, making its cleavage sites dynamically exposed.

General

2) The proposed mechanism is mainly based on that force may stablize the bound ABD in the higher affinity conformation where the H0 and H1 helices are dissociated from H2-H5. Previous study by Huang et al., (2017) revealed that, when the force applied to vinculin ABD is toward the minus end of F-actin, a switch from catch-bond to slip-bond occurred at ~ 8-10 pN. Based on the proposed mechanism in this work, the force needed to stabilize the ABD in the H1 displaced state should be in a similar level. This again suggests that H1 associates with H2-H5 with significant interaction energy when the ABD is bound on F-actin, which requires a few pN forces to dissociate.

3) The study indicated role of the long C-terminal extension (CTE), residues 844-906, which follows H5. According to the structure, the disordered CTE past 860 forms an extended peptide that interacts with F-actin. Previous study by Huang et al., (2017) also revealed that, when the force applied to vinculin ABD is toward the plus end of F-actin, although the mechanical stability of vinculin ABD on F-actin is significantly weaker than that toward the minus end, a switch from catch-bond to slip-bond was still observed at ~ 8-10 pN forces. Apparently, this plus-end catch-to-slip switch behavior cannot be explained by the force-dependent dissociation of the H1 helix from H2-h5. However, I think it might be explained by the presence of the pre-extended CTE peptide on F-actin.

4) Several recent studies (Yuan et al., 2017; Guo et al., 2018; 2020) suggested that if unfolding of a domain or a rupturing of a biomolecular complex follows a transition pathway where a pre-extended peptide is peeled away from the remaining folded core in a shearing force geometry, a catch-to-slip switch behavior can be expected. This phenomenon is related to the highly flexible nature of the peptide, leading to a negative transition distance within certain tension range up to a value of f_s (therefore catch bond) which then switches to positive values (therefore slip bond) when tension exceeds f_s. The more the peptide is pre-extended in the native structure, the larger the switching tension f_s. Based on this physical mechanism, if the pre-extended CTE is the last anchoring site of the ABD stretched toward the plus end of F-actin, a catch-to-slip switch behavior is predicted. If the authors think the plus-end catch-to-slip switch behavior an interesting phenomenon, it will be good to add several sentences on this point in the Discussion section.

5) Citations of published structures are often missing both in the manuscript and figure legends. Both the original publication and PDB accession number should be stated.

Introduction

6) "In the optical trap data" – needs a citation.

7) "Comparison of the crystal structure and the actin bound form of the complete aE-catenin ABD, as well as the structures of the vinculin ABD free or bound to actin (Bakolitsa et al., 2004; Bakolitsa et al., 1999; Borgon et al., 2004; Mei et al., 2020), provides an explanation for the weak to strong actin-binding transition, and biochemical and mutational data support this model." – It is unclear which citation belongs to which structure and what is presented in this paper.

Results section

8) "[…] we obtained a three-dimensional cryo-EM reconstruction of actin filaments bound to a truncated aE-catenin ABD (residues 671-906) […] This construct deletes the first half of, and thereby destabilizes, the short H0, and binds 4.5x more strongly than the complete ABD" – Why was this construct chosen? Was the background too high when using the full ABD and a higher concentration? What is the affinity of the full-length protein? Since this construct results in a sole population of the strong binding state, and thus levers out the catch bond mechanism and possibility to solve the structure of both states (weak and strong binding), this point should be discussed in more detail in the manuscript. Moreover, the authors themselves note that Mei et al. used the full ABD, arising the question why they didn't.

9) "[…] we observed either bare actin filaments or stretches of filaments continuously bound by aE-ABD. Using the same 10 μM concentration, we were unable to observe binding of the complete ABD to actin filaments in the electron microscope." – An additional SI figure showing examples for bare actin and decorated stretches for both cases would be helpful. Also, it's not clear to me, if all this was done using cryo-EM and if decoration was judge by eye on micrograph level or if for example layer lines were used.

10) "Indeed, a comparable structure using an ABD construct spanning residues 664-906 was produced using a higher concentration of ABD (Mei et al., 2020)." – Stating the concentration used by Mei et al., and the corresponding affinity would be helpful.

11) "The visible portion of the aE-catenin ABD starts at residue 699 and ends at 871; 6 residues in the loop connecting H4 and H5, and 8 residues in the connection between H5 and the rest of the C-terminal extension also could not be modeled." – readers without cryo-EM background require an explanation what "visible portion" implies.

12) A short mentioning of the actin isoform and its nucleotide state/ overall buffer condition is necessary to follow the discussion of the D-loop state.

13) " […] modeling suggests that the 'open' D-loop conformation that has been associated with the ATP-bound form of bare actin filaments (Merino et al., 2018) would clash with the bound aE-catenin" – This is interesting. Does aE-catenin prefer a certain nucleotide state of actin?

14) "Moreover, the refinement procedure used to generate high-resolution structures from cryo-EM images selects and enforces a single conformation" – please specify that helical refinement was used and add a short explanation for non-expert readers, how a single conformation is enforced.

15) I am missing a few aspects in the discussion of the binding site/mode: As aE-catenin contacts two actin subunits, does it also stabilize F-actin? Does the binding site overlap with the binding site of other actin binding proteins? Do they colocalize in the cell? Is a full-length crystal structure available? Or data about how the remainder of catenin packs relative to the ABD? Would it introduce clashes/ additional interactions between neighboring molecules? Maybe the authors could add a "sketch" to one of the figures.

16) "[…] N-terminus through the last turn of H1 is disordered (Figure 3A). We note that an F-actin-bound aE-catenin ABD structure has been reported recently for the complete ABD (664-906) (Mei et al., 2020), and the same residues are disordered, demonstrating that the truncation of H0 in our construct has no influence on the actin bound structure." – Can the authord speculate on the function/stabilization of the disordered part in the full-length protein?

17) "This repositions an aromatic cluster formed by conserved residues W705 (H1), Y837 (H5), and W859 (CTE), and shifts the CTE upward (Figure 3C)." Please highlight these residues in Figure 2—figure supplement 2B.

18) "The CTE past 860, which is disordered in the isolated structure, forms an extended peptide that interacts with actin." Interactions with actin need to be specified/ described in more detail. A 2D interaction plot might be helpful.

19) "[...] lowers the affinity for actin approximately 10-fold (Table 1)." Reference to wrong table.

20) "Chen et al., (et al.2015) found several point mutants that severely weakened binding, including I792A. K842A eliminates contacts with actin residues H87 and Y91; and K866A eliminates side chain and main chain contacts with actin residues R28 and V30 (Figure 2D)." Reference of I792 to structure is missing, no figure showing the interactions of K842 and R28.

21. "Surprisingly, removal of residues 869-906 eliminated detectable F-actin binding, even though none of these residues interact with actin in our structure." Can the authors offer a hypothesis for what the function of residues 869-883 could be?

22) "Moreover, although relatively few of the actin-contacting residues are conserved in vinculin (notably, those in the C-terminal portion of H4), the vinculin ABD undergoes a similar structural transition upon binding to actin" a superposition of both structures in a supplementary figure would help the reader to follow this point.

23) "We cannot assess whether the weak state corresponds precisely to the solution conformation, or if it is intermediate between the experimentally determined unbound and actin-bound structures." I agree that it's impossible with the author's construct/ data, but shouldn't it be possible to get both states with a longer construct that still undergoes the conformational change from weak to strong binding?

24) Discussion section

25) What follow-up experiments could help to clarify the mechanism? Structure with longer construct? Interesting target for MD simulations?

Materials and methods section

26) Values in Manuscript/ Methods and Table 1 do not match! Resolution 3.8 vs 3.6 Å, different symmetry values? All values need to be checked and corrected!

27) "The data were analyzed in the program GraphPad Prism (GraphPad Software, La Jolla California USA) and fitted with a 'single binding with Hill coefficient' model. The data were analyzed in the program GraphPad Prism and fitted using the 'specific binding with Hill slope' model, with the exception of the aE-catenin 671-906 W859A mutant." Partially duplicated sentence.

28) "The defocus was estimated either using Gctf (Zhang, 2016) or CTFFIND4 (Rohou and Grigorieff, 2015)" Why were two different programs used to estimate the defocus of a single data set? Which criterion was used to decide which result to use?

29) Is there a reason why particles were not polished and/or CTF refined? Did the authors try 3D classification (focused, possibly in combination with signal subtraction)?

30) " The reconstruction was then symmetrized within pyCoAn using the refined helical parameters. Real space refinement was performed with Phenix (Afonine et al., 2018). Local resolution estimates were calculated with ResMap (Kucukelbir et al., 2014) and RELION3." Remove sentence about real space refinement here (discussed in detail in the next paragraph)

31) Missing version numbers for some software packages including Phenix, CTFFIND, GCTF and others.

32) Cryo-EM resolutions should only be stated with one decimal digit.

Tables

33) Table 1: Please add exposure time, number of non-hydrogen atoms, Rotamer statistics and cross correlation values. A consistent style should be used for numbers i.e. 4,769 mics not 4769. Cryo-EM resolutions should only be stated with one decimal digit.

Figures and Legends

34) Mismatch of capitalization of panel lettering in manuscript and figures i.e. Figure 1a vs 1A.

35. Figure 1: Showing cadherins from different cells in different shades of green would help to understand the figure. Figure legend: "connection to the actin cytoskeleton", "Crystal structure of the aE-catenin ABD" PDB accession numbers should be stated. Highlights of binding sites in B are not explained in the figure legends.

36) Figure 2: Typo in figure caption " aE-catenin ABD-F-catenin complex". Boxes that show regions enlarged in C) and D) should be added in B. Why is G46 shown in ball representation? Why are labels shown in red and not in magenta to match the structure? Sticking to one color code helps the reader. Label for L803 hardly readable, suggest white instead of yellow background for residue labels.

37) Figure 2—figure supplement 1A) missing scale bar, B) FSC curve needs to be shown up to 0.5 (Nyquist) and y axis >1, as not only the value is of importance but also the curve itself especially the oscillation around zero close to Nyquist. C) Color range for local resolution is not meaningful and should be adjusted.

38) Figure 2—figure supplement 2A) Color code not explained properly in legend, important residues i.e. hydrophobic cluster should be highlighted.

39) Figure 3: The color code in the legend does not completely correspond to the figure. Missing verb in "The upper panel packing interactions of H0 residues 672, M673 and L676 with H5 residues V809, G811 and A815, […]". Missing citation and PDB accession number of the unbound state. Side chains are hardly visible in C) View needs to be changed / another one added to emphasize side chains and the movement of the CTE. The order from left to right to bottom is confusing.

40) Figure 4: Enlarge A, Labels in B) not legible. Consider a white outline/background.

41) Figure 5: "This revealed only minor clashes with actin at the N-terminus of H5, and modeling suggests these can be alleviated by changes of side chain rotamers (Figure 5A, left)." Neither the clashes or proposed side chain rotations are apparent. An extra view and especially a movie would help.

42) The manuscript might benefit from the inclusion of movies. For example, to highlight the binding site of aE-catenin and show close-up views of interactions with F-actin and between ABDs. Another movie could show a morph of the unbound and bound structure of aE-catenin, and an animation of the catch bond model.

---

## [Author Response]

Essential revisions:Abstract1) In the Abstract, the authors mentioned "Upon binding to actin, the first helix of the bundle dissociates and the remaining four helices and connecting loops rearrange to form the interface with actin." This conclusion is suggested from the higher proteolytic sensitivity of the H0-deleted ABD in the presence of F-actin, instead of a direct result. However, if the binding to F-actin is enough to dissociate H1, why do the authors propose that there are two populations of the bound ABDs, a weaker bound state where the H1 is not dissociated and a stronger bound where the H1 is displaced, which equilibrium is shifted by the force applied to the ABD? To allow the force play a significant role, it seems to me that in the absence of force, the auto-inhibited weaker species should be the prevalent one, so that the force can modulate the binding by releasing this auto-inhibition. Perhaps it is more reasonable to think that the observed higher proteolytic sensitivity of the H0-deleted ABD in the presence of F-actin indicates that the H1 helix becomes more dynamic, instead of being dissociated, making its cleavage sites dynamically exposed.

The existence of two bound states comes directly from optical trap data (Buckley et al., 2014, Huang et al., 2017), which showed that for any given force there is a bi-exponential distribution of lifetimes that can be explained only if there are two bound states. The best model for these data was that either state could dissociate from actin (with different kinetics), and that there is an equilibrium between the weak and strong-bound states that is altered by force. This is the underlying premise of the paper and is described in the introduction. As we discuss in the present paper, conformational change in the ABD is coupled to the binding energy to actin, i.e. in the presence of actin the barrier to the conformational change is lower.

We phrased the cited sentence badly – the stable (strong state) form is what is rearranged, but we are proposing that the weak state corresponds to a structure close to that of non-H1 dissociated form. We have modified the abstract to clarify the origin of the two bound states model. We have also further clarified the role of force on the equilibrium in subsection “Insights into catch bond mechanism”.

The proteolysis experiments are performed on a minutes time scale, so although both forms are sampled, clipping the helix will effectively shift the equilibrium, so that over the time of the experiment it is completely lost.

General2) The proposed mechanism is mainly based on that force may stablize the bound ABD in the higher affinity conformation where the H0 and H1 helices are dissociated from H2-H5. Previous study by Huang et al., (2017) revealed that, when the force applied to vinculin ABD is toward the minus end of F-actin, a switch from catch-bond to slip-bond occurred at ~ 8-10 pN. Based on the proposed mechanism in this work, the force needed to stabilize the ABD in the H1 displaced state should be in a similar level. This again suggests that H1 associates with H2-H5 with significant interaction energy when the ABD is bound on F-actin, which requires a few pN forces to dissociate.

We agree with the reviewer. As noted in the response to point 1, both weak and strong state can bind to actin and force shifts the equilibrium between them, and in the weak state we propose that H1 is associated with H2-H5, albeit with possibly small changes relative to the unbound structure (see text). The slip behavior at 8 pN is due to removing the entire bundle from actin. We note that a-catenin, although homologous, is not vinculin, so there is no a priori reason to believe that the detailed force dependence would be the same. Nonetheless, we did observe a transition to slip at a similar force value for the cadherin/b-catenin/a-catenin complex in Buckley et al., 2014.

3) The study indicated role of the long C-terminal extension (CTE), residues 844-906, which follows H5. According to the structure, the disordered CTE past 860 forms an extended peptide that interacts with F-actin. Previous study by Huang et al., (2017) also revealed that, when the force applied to vinculin ABD is toward the plus end of F-actin, although the mechanical stability of vinculin ABD on F-actin is significantly weaker than that toward the minus end, a switch from catch-bond to slip-bond was still observed at ~ 8-10 pN forces. Apparently, this plus-end catch-to-slip switch behavior cannot be explained by the force-dependent dissociation of the H1 helix from H2-h5. However, I think it might be explained by the presence of the pre-extended CTE peptide on F-actin.

The kinetics observed in the optical trap were described by a modified Bell model in which the transition state energy depends on the force acting over the distance from the ground to transition state (see Buckley et al., 2014 and Huang et al., 2017). As the reviewer notes, the force dependence doesn’t change depending on direction, only the bound lifetime, implying a different distance to the transition state in the two directions. A geometric model can account for force stabilizing the strongly bound state regardless of whether the force is directed toward the plus or minus end of the actin filament (see Huang et al.,). Although at present we do not have a detailed molecular model for the transition between weak and strong states, as we tried to illustrate in Figure 6 and the short animation (Videos 1 and 2), the force vector experienced by a-catenin or vinculin is not necessarily aligned with the actin filament, and will depend in part on the geometry of how the molecule is tethered. In addition, the component of the force needed to remove H1 (and H0 in the case of a-catenin) is unlikely to be precisely coincident with the helix axis of actin. Thus, the projection of force onto the reaction coordinate can have the same sign regardless of whether the actin filament is being pulled on from its (-) or (+) end.

4) Several recent studies (Yuan et al., 2017; Guo et al., 2018; 2020) suggested that if unfolding of a domain or a rupturing of a biomolecular complex follows a transition pathway where a pre-extended peptide is peeled away from the remaining folded core in a shearing force geometry, a catch-to-slip switch behavior can be expected. This phenomenon is related to the highly flexible nature of the peptide, leading to a negative transition distance within certain tension range up to a value of f_s (therefore catch bond) which then switches to positive values (therefore slip bond) when tension exceeds f_s. The more the peptide is pre-extended in the native structure, the larger the switching tension f_s. Based on this physical mechanism, if the pre-extended CTE is the last anchoring site of the ABD stretched toward the plus end of F-actin, a catch-to-slip switch behavior is predicted. If the authors think the plus-end catch-to-slip switch behavior an interesting phenomenon, it will be good to add several sentences on this point in the Discussion section.

Thank you for pointing out these papers and model. The idea that the disorder-order transition of the CTE contributes to the behavior is extremely interesting, and we have added a brief discussion of this idea. Another effect that may contribute to the transition from the strong to weakly bound state, which we did not address in the original manuscript, is that H1 (and H0) may unfold (as suggested by the proteolysis data) when they are detached from the rest of the ABD. If so, they would provide a flexible linker that extends in the direction of force and contributes to the catch bond behavior. More broadly, we plan to do further modeling with MD simulations to assess whether the data can be modeled in a way that provides physical parameters, but that is beyond the scope of this work. Crucially, the situation here is complicated by the fact that the position of the CTE is coupled to the order or disorder of H1. This could mean that the order-to-disorder of the CTE in going from strong to weak state contributes to the behavior, but because this change is concerted with that of H1, its effect can’t be sorted out cleanly. Finally, in our understanding, the specific model presented in the cited papers predict mono-exponential decay at any given force, which is not what was observed for either a-catenin or vinculin.

5) Citations of published structures are often missing both in the manuscript and figure legends. Both the original publication and PDB accession number should be stated.

Citations and PDB accession number were added in the appropriate places.

Introduction6) "In the optical trap data" needs a citation.

(Buckley et al., 2014; Huang et al., 2017) were both cited in this sentence

7) "Comparison of the crystal structure and the actin bound form of the complete aE-catenin ABD, as well as the structures of the vinculin ABD free or bound to actin (Bakolitsa et al., 2004; Bakolitsa et al., 1999; Borgon et al., 2004; Mei et al., 2020), provides an explanation for the weak to strong actin-binding transition, and biochemical and mutational data support this model." It is unclear which citation belongs to which structure and what is presented in this paper.

We have made these corrections (Introduction).

Results section8) "[…] we obtained a three-dimensional cryo-EM reconstruction of actin filaments bound to a truncated aE-catenin ABD (residues 671-906) […] This construct deletes the first half of, and thereby destabilizes, the short H0, and binds 4.5x more strongly than the complete ABD" Why was this construct chosen? Was the background too high when using the full ABD and a higher concentration? What is the affinity of the full-length protein? Since this construct results in a sole population of the strong binding state, and thus levers out the catch bond mechanism and possibility to solve the structure of both states (weak and strong binding), this point should be discussed in more detail in the manuscript. Moreover, the authors themselves note that Mei et al. used the full ABD, arising the question why they didn't.

In our hands, the full-length ABD construct produced very sparse binding to actin filaments in the concentration ranges we used. From our earlier studies (Hansen et al., 2013) we already knew that the 671-906 construct results in excellent decoration for helical averaging, making it an obvious choice for improving the reconstruction without increasing the background too much. Mei et al., (2020) used a 2x higher concentration and a somewhat different protocol for preparing the cryo specimens of the full ABD construct, but crucially, they observed the same structure, i.e., starting at 699, with H0 and H1 dissociated and the H2-H5-CTE region rearranged. Thus, both constructs populate the presumed strong state in the reconstructions. It is likely still in an equilibrium but this equilibrium is shifted towards the strong binding state for the 671-906 construct relative to the full-length ABD, reflected in its ~4x higher affinity (Table 2). We cannot rule out that the conditions of the EM specimen preparation may affect the equilibrium and shift it further to the strong state. We have clarified these differences in the text (subsection “Structure of aE-catenin ABD bound to F-actin”).

Regarding the full-length a-catenin, it binds with about the same K_D_ as the ABD alone (but this is a dimer, so we don’t have a direct comparison to monomeric ABD), but unlike the ABD it does not bind with strong cooperativity (Hansen et al., 2013). Consistently, we have only observed sparse decoration with the full-length protein, preventing reconstructions.

9) "[…] we observed either bare actin filaments or stretches of filaments continuously bound by aE-ABD. Using the same 10 μM concentration, we were unable to observe binding of the complete ABD to actin filaments in the electron microscope." An additional SI figure showing examples for bare actin and decorated stretches for both cases would be helpful. Also, it's not clear to me, if all this was done using cryo-EM and if decoration was judge by eye on micrograph level or if for example layer lines were used.

We added representative micrographs for both full-length and truncated ABD data with markings for bare and decorated actin filaments to Figure 2—figure supplement 1 (A and B). All of the analysis was done by cryo-EM, including initial screening. Decoration, which is readily apparent in the micrographs was judged initially by visual inspection using high-defocus data from the 120 keV cryo-TEM microscope while screening conditions. Decoration and the noted cooperativity was confirmed in the data set computationally during data processing. We have now noted these in the Materials and methods section.

10) "Indeed, a comparable structure using an ABD construct spanning residues 664-906 was produced using a higher concentration of ABD (Mei et al., 2020)." Stating the concentration used by Mei et al., and the corresponding affinity would be helpful.

They used 20 mM, i.e. 2x higher. Our measured affinity for our full-length ABD construct (666-906) was 8.5 mM. We have now noted these in the text (subsection “Structure of aE-catenin ABD bound to F-actin”).

11) "The visible portion of the aE-catenin ABD starts at residue 699 and ends at 871; 6 residues in the loop connecting H4 and H5, and 8 residues in the connection between H5 and the rest of the C-terminal extension also could not be modeled." Readers without cryo-EM background require an explanation what "visible portion" implies.

“Visible portion” was perhaps a poor choice of words. We rewrote this section as follows (subsection “Structure of aE-catenin ABD bound to F-actin”):

“For the aE-catenin ABD, there was no detectable density for residues 671-698, or from 872-906. In addition, 6 residues in the loop connecting H4 and H5, and 8 residues in the connection between H5 and the rest of the C-terminal extension also could not be modeled.”

12) A short mentioning of the actin isoform and its nucleotide state/ overall buffer condition is necessary to follow the discussion of the D-loop state.

We now note that ADP actin is used, with its preparation given in the Materials and methods section.

13) " […] modeling suggests that the 'open' D-loop conformation that has been associated with the ATP-bound form of bare actin filaments (Merino et al., 2018) would clash with the bound aE-catenin" This is interesting. Does aE-catenin prefer a certain nucleotide state of actin?

There are no data available on the nucleotide preference. Also, in retrospect we gave a too-definitive statement. In fact, while the ADP state of vertebrate actin appears to be tightly coupled with the ‘closed’ conformation of the D-loop, other nucleotide states, including ATP, clearly show an equilibrium between the ‘closed’ and the alternative ‘open’ D-loop conformation. We replaced the original statement with (this in the text (subsection “Structure of aE-catenin ABD bound to F-actin”)):

“Modeling suggests that the alternative ‘open’ D-loop conformation that occurs in equilibrium with the ‘closed’ conformation in other nucleotide states (Merino et al., 2019), may clash with the bound aE-catenin.”

14) "Moreover, the refinement procedure used to generate high-resolution structures from cryo-EM images selects and enforces a single conformation" please specify that helical refinement was used and add a short explanation for non-expert readers, how a single conformation is enforced.

We have now explained this in detail in the Materials and methods section, and refer to that in the main text.

15) I am missing a few aspects in the discussion of the binding site/mode: As aE-catenin contacts two actin subunits, does it also stabilize F-actin? Does the binding site overlap with the binding site of other actin binding proteins? Do they colocalize in the cell? Is a full-length crystal structure available? Or data about how the remainder of catenin packs relative to the ABD? Would it introduce clashes/ additional interactions between neighboring molecules? Maybe the authors could add a "sketch" to one of the figures.

We have no direct evidence that aE-catenin stabilizes F-actin. More broadly, while interesting, we feel that discussion of overlapping/competing sites and subcellular localization would distract from the focus of this paper. The binding site does overlap other binding sites, in particular that of Arp2/3, and we have published that aE-catenin competes with Arp2/3 for binding (Drees et al., 2005). We are not aware of any other such competition studies. Regarding full-length aE-catenin, small angle x-ray scattering data indicate that the ABD is flexibly linked to the rest of the molecule (Ishiyama et al., 2013; Nicholl et al., 2018; Terekhova et al., 2019), so we cannot provide a sensible model showing the full protein. We previously published that full length aE-catenin by itself does not show strong cooperativity of binding vs. the ABD (Hansen et al., 2013), which could be due to steric interference of neighboring molecules. However, the relevant complex for tension is when aE-catenin is associated with cadherin/b-catenin, and there are no data that speak directly to this point.

16) "[…] N-terminus through the last turn of H1 is disordered (Figure 3A). We note that an F-actin-bound aE-catenin ABD structure has been reported recently for the complete ABD (664-906) (Mei et al., 2020), and the same residues are disordered, demonstrating that the truncation of H0 in our construct has no influence on the actin bound structure." Can the authord speculate on the function/stabilization of the disordered part in the full-length protein?

We have added the following to the Discussion section:

“Although it is not apparently integral to the catch bond behavior, we speculate that the conserved H0 of a-catenin serves to tune the stability and force response of the ABD; in the disordered state bound to actin, it will provide additional flexible elements that may affect the detailed catch bond behavior.”

17) "This repositions an aromatic cluster formed by conserved residues W705 (H1), Y837 (H5), and W859 (CTE), and shifts the CTE upward (Figure 3C)." Please highlight these residues in Figure 2—figure supplement 2B.

Added to the alignment figure as requested (now Figure 2—figure supplement 5).

18) "The CTE past 860, which is disordered in the isolated structure, forms an extended peptide that interacts with actin." Interactions with actin need to be specified/ described in more detail. A 2D interaction plot might be helpful.

Added details to the text (his in the text (subsection “Structure of aE-catenin ABD bound to F-actin”)) and a new figure (Figure 2—figure supplement 3A).

19) "[...] lowers the affinity for actin approximately 10-fold (Table 1)." Reference to wrong table.

Corrected.

20) "Chen et al., (2015) found several point mutants that severely weakened binding, including I792A. K842A eliminates contacts with actin residues H87 and Y91; and K866A eliminates side chain and main chain contacts with actin residues R28 and V30 (Figure 2D)." Reference of I792 to structure is missing, no figure showing the interactions of K842 and R28.

I792 is described in Figure 2C and referenced in the preceding part of the paragraph. Added additional Figure 2—figure supplement 3B and C to show interactions of K842 as well as K797.

21) "Surprisingly, removal of residues 869-906 eliminated detectable F-actin binding, even though none of these residues interact with actin in our structure." Can the authors offer a hypothesis for what the function of residues 869-883 could be?

We do not have a good hypothesis; we think that there may be a dynamic interaction that averages out in the structure. Mei et al. propose that 872-906 mediate additional binding when actin itself is subject to tension. It could be that 869-871 form contacts that we cannot assign due to the limited resolution of our structure. We now discuss this in the text (subsection “Structure of aE-catenin ABD bound to F-actin”):

(Mei et al., 2020) proposed that these C-terminal residues absent in the structure may mediate a small increase in affinity when actin is placed under tension. However, deleting these residues weakens the affinity (albeit slightly) in a solution assay, which indicates that they have a role independent of tension. It is possible that these are highly dynamic interactions that are not sufficiently stable to be visualized in the cryo-EM structure.

22) "Moreover, although relatively few of the actin-contacting residues are conserved in vinculin (notably, those in the C-terminal portion of H4), the vinculin ABD undergoes a similar structural transition upon binding to actin" a superposition of both structures in a supplementary figure would help the reader to follow this point.

We have added comparisons of the free and actin-bound vinculin and aE-catenin structures as Figure 3—figure supplement 2. We used the 8.5 Å resolution vinculin structure from Kim et al., 2016 as the higher resolution structure from Mei et al., (2020) is not available.

23) "We cannot assess whether the weak state corresponds precisely to the solution conformation, or if it is intermediate between the experimentally determined unbound and actin-bound structures." I agree that it's impossible with the author's construct/ data, but shouldn't it be possible to get both states with a longer construct that still undergoes the conformational change from weak to strong binding?

No, because it seems to equilibrate to the state seen here, and the longer construct is even weaker (see above regarding the preparation of samples). We also note that Mei et al., 2020 do not report any heterogeneity in the structures.

24) Discussion section25) What follow-up experiments could help to clarify the mechanism? Structure with longer construct? Interesting target for MD simulations?

We now note that MD simulations and more optical trap experiments are being done in order to better define the mechanism (Discussion section).

Materials and methods section26) Values in Manuscript/ Materials and methods section and Table 1 do not match! Resolution 3.8 vs 3.6 Å, different symmetry values? All values need to be checked and corrected!

Values have been checked and corrected in the text.

27) "The data were analyzed in the program GraphPad Prism (GraphPad Software, La Jolla California USA) and fitted with a 'single binding with Hill coefficient' model. The data were analyzed in the program GraphPad Prism and fitted using the 'specific binding with Hill slope' model, with the exception of the aE-catenin 671-906 W859A mutant." Partially duplicated sentence.

Corrected in the text.

28) "The defocus was estimated either using Gctf (Zhang, 2016) or CTFFIND4 (Rohou and Grigorieff, 2015)" – Why were two different programs used to estimate the defocus of a single data set? Which criterion was used to decide which result to use?

The data were collected in several separate imaging sessions. Initially, we used CTFFIND4. We later switched to Gctf, which gives essentially the same results as CTFFIND4 but is substantially faster because it is GPU accelerated. This is now described in the Materials and methods section.

29) Is there a reason why particles were not polished and/or CTF refined? Did the authors try 3D classification (focused, possibly in combination with signal subtraction)?

The CTF was, in fact, refined. Particle polishing was not used because anisotropic motion correction and dose weighting was already applied within motioncor2. We reflected these facts in the updated Materials and methods section.

30) "The reconstruction was then symmetrized within pyCoAn using the refined helical parameters. Real space refinement was performed with Phenix (Afonine et al., 2018). Local resolution estimates were calculated with ResMap (Kucukelbir et al., 2014) and RELION3." Remove sentence about real space refinement here (discussed in detail in the next paragraph).

Removed from the text.

31) Missing version numbers for some software packages including Phenix, CTFFIND, GCTF and others.

Version numbers were added.

32) Cryo-EM resolutions should only be stated with one decimal digit.

Revised in the text.

Tables33) Table 1: Please add exposure time, number of non-hydrogen atoms, Rotamer statistics and cross correlation values. A consistent style should be used for numbers i.e. 4,769 mics not 4769. Cryo-EM resolutions should only be stated with one decimal digit.

The values were added to the Table.

Figures and Legends34) Mismatch of capitalization of panel lettering in manuscript and figures i.e. Figure 1A vs 1A.

Capitalization is now consistent with panels.

35. Figure 1: Showing cadherins from different cells in different shades of green would help to understand the figure. Figure legend: "connection to the actin cytoskeleton", "Crystal structure of the aE-catenin ABD" PDB accession numbers should be stated. Highlights of binding sites in B are not explained in the figure legends.

Corrected.

36) Figure 2: Typo in figure caption " aE-catenin ABD-F-catenin complex". Boxes that show regions enlarged in C) and D) should be added in B. Why is G46 shown in ball representation? Why are labels shown in red and not in magenta to match the structure? Sticking to one color code helps the reader. Label for L803 hardly readable, suggest white instead of yellow background for residue labels.

G46 is shown as a ball because showing side chains only on the cartoon representation does not show α-carbons. Labels are color matched to structure; the yellow background makes them more readable but may make the color look slightly different. We prefer to use this scheme.

37) Figure 2—figure supplement 1A) missing scale bar, B) FSC curve needs to be shown up to 0.5 (Nyquist) and y axis >1, as not only the value is of importance but also the curve itself especially the oscillation around zero close to Nyquist. C) Color range for local resolution is not meaningful and should be adjusted.

Scale bars were added to A and B. FSC curve was extended. The color range for local resolution was adjusted to show a narrower resolution range.

38) Figure 2—figure supplement 2A) Color code not explained properly in legend, important residues i.e. hydrophobic cluster should be highlighted.

Corrected in the text.

39) Figure 3: The color code in the legend does not completely correspond to the figure. Missing verb in "The upper panel packing interactions of H0 residues 672, M673 and L676 with H5 residues V809, G811 and A815, […]". Missing citation and PDB accession number of the unbound state. Side chains are hardly visible in C) View needs to be changed / another one added to emphasize side chains and the movement of the CTE. The order from left to right to bottom is confusing.

The changes in the CTE due to repacking of the aromatic cluster are now in a separate figure (Figure 3 – figure supplement 1), where we added an overlay of the free and bound structures to better show the change in the CTE position.

40) Figure 4: Enlarge A, Labels in B) not legible. Consider a white outline/background.

Enlarged and outlined labels in B.

41) Figure 5: "This revealed only minor clashes with actin at the N-terminus of H5, and modeling suggests these can be alleviated by changes of side chain rotamers (Figure 5A, left)." Neither the clashes or proposed side chain rotations are apparent. An extra view and especially a movie would help.

We have added a Figure 5—figure supplement 1 to show the major clash with actin and how small movements of H1 and the start of H5 would relieve them. We also clarified the text (subsection “Insights into catch bond mechanism”) to better explain the model.

42) The manuscript might benefit from the inclusion of movies. For example, to highlight the binding site of aE-catenin and show close-up views of interactions with F-actin and between ABDs. Another movie could show a morph of the unbound and bound structure of aE-catenin, and an animation of the catch bond model.

We have added animated versions of Figure 6 in order to intuitively capture our model (Video 1 and Video 2).